# GCN5L1 modulates cross-talk between mitochondria and cell signaling to regulate FoxO1 stability and gluconeogenesis

Lingdi Wang[1], Iain Scott[2], Lu Zhu [3], Kaiyuan Wu[1], Kim Han[1], Yong Chen[4], Marjan Gucek[4] & Michael N. Sack[1]

The mitochondrial enriched GCN5-like 1 (GCN5L1) protein has been shown to modulate mitochondrial protein acetylation, mitochondrial content and mitochondrial retrograde signaling. Here we show that hepatic GCN5L1 ablation reduces fasting glucose levels and blunts hepatic gluconeogenesis without affecting systemic glucose tolerance. *PEPCK* and *G6Pase* transcript levels are downregulated in hepatocytes from GCN5L1 liver specific knockout mice and their upstream regulator, FoxO1 protein levels are decreased via proteasome-dependent degradation and via reactive oxygen species mediated ERK-1/2 phosphorylation. ERK inhibition restores FoxO1, gluconeogenic enzyme expression and glucose production. Reconstitution of mitochondrial-targeted GCN5L1 blunts mitochondrial ROS, ERK activation and increases FoxO1, gluconeogenic enzyme expression and hepatocyte glucose production. We suggest that mitochondrial GCN5L1 modulates post-translational control of FoxO1, regulates gluconeogenesis and controls metabolic pathways via mitochondrial ROS mediated ERK activation. Exploring mechanisms underpinning GCN5L1 mediated ROS signaling may expand our understanding of the role of mitochondria in gluconeogenesis control.

[1] Cardiovascular and Pulmonary Branch, National Heart, Lung and Blood Institute, NIH, Bethesda, MD 20892, USA. [2] Cardiology Division, Department of Medicine, University of Pittsburgh Medical Center, Pittsburgh, PA 15261, USA. [3] Molecular Signaling Section, Laboratory of Bioorganic Chemistry, National Institute of Diabetes and Digestive and Kidney Diseases, Bethesda, MD 20892, USA. [4] Proteomics Core Facility, National Heart, Lung and Blood Institute, NIH, Bethesda, MD 20892, USA. Correspondence and requests for materials should be addressed to M.N.S. (email: sackm@nih.gov)

GCN5L1 shares sequence homology to the nuclear acetyltransferase GCN5, but is enriched within mitochondria and the cytosol[1, 2]. Although GCN5L1 possesses prokaryote-conserved acetyltransferase substrate and acetyl-CoA-binding regions, it lacks an acetyltransferase catalytic domain[1]. The absence of the catalytic domain is consistent with the inability of GCN5L1 alone to initiate in-vitro protein acetylation. However its depletion blunted, and its reconstitution restored mitochondrial protein acetylation[1, 3]. In parallel the stable partial knockdown of GCN5L1 resulted in decreased mitochondrial protein acetylation with the concurrent induction of selective mitophagy and the reduction in mitochondrial content[3]. In addition the complete ablation of GCN5L1 in MEF cells exhibited increased mitochondrial turnover with the parallel induction of mitophagy and mitochondrial biogenesis thereby retaining mitochondrial content[4]. These counter-regulatory programs were found to be dependent on the concurrent upregulation of lysosomal (TFEB) and mitochondrial (PGC-1α) master regulators in response to GCN5L1 depletion[4]. Taken together, these data show that GCN5L1 plays an important role in mitochondrial protein acetylation and that its disruption induces mitochondrial to nuclear signaling[5, 6]. At the same time, GCN5L1 has also been named BLOC1S1, and had been found to be a component of the cytosolic biogenesis of lysosome-related organelles complex-1 (BLOC1), where it plays a role in protein trafficking to the endosomal–lysosomal complex[2]. These cell compartment specific roles of GCN5L1 have not been reconciled and the molecular and biochemical mechanisms underpinning the function of this protein remain poorly understood.

To begin to address some of these issues, GCN5L1 knockout mouse was generated but found to be embryonic lethal[2, 4]. To explore the function of GCN5L1 in-vivo, and given that the GCN5L1 protein is highly regulated in the liver by feeding and fasting[3], we generated a liver-specific knockout (LKO) mouse. These mice were viable and appeared healthy, and in parallel with prior cell culture studies[3, 4, 7], mitochondria extracted from primary hepatocytes of GCN5L1 LKO mice showed mitochondrial-restricted reduction in protein acetylation. Initial phenotypic screening the GCN5L1 LKO mice showed evidence of fasting hypoglycemia and impaired glucose production in response to a pyruvate challenge.

Given that gluconeogenesis is essential for the maintenance of blood glucose levels during fasting and its dysregulation augments hyperglycemia in Type 2 diabetes mellitus we began to explore the role of GCN5L1 in regulating this glucose production pathway. The liver is the major site of gluconeogenesis and the signaling and transcriptional programs controlling hepatic glucose production are well characterized[8–11]. Furthermore, additional studies have begun to define the post-translational control of gluconeogenic enzymes under distinct nutrient load dependent conditions[12, 13] and levels of acetyl-CoA similarly control the rate of gluconeogenesis from yeast to mammals[14, 15].

In this study, we found that the knockout of GCN5L1 in the liver results in a modest basal reduction in mitochondrial protein acetylation. Furthermore, we found that fasting glucose production is reduced in KO mice and these mice have a reduced ability to generate glucose in response to a pyruvate challenge. In addition, primary LKO hepatocytes exhibit reduced baseline and glucagon-induced glucose production and downregulation of gluconeogenesis regulatory transcripts and proteins including PEPCK and G6Pase. This program is blunted, in part, due to the diminished stability of the gluconeogenesis transactivator FoxO1.

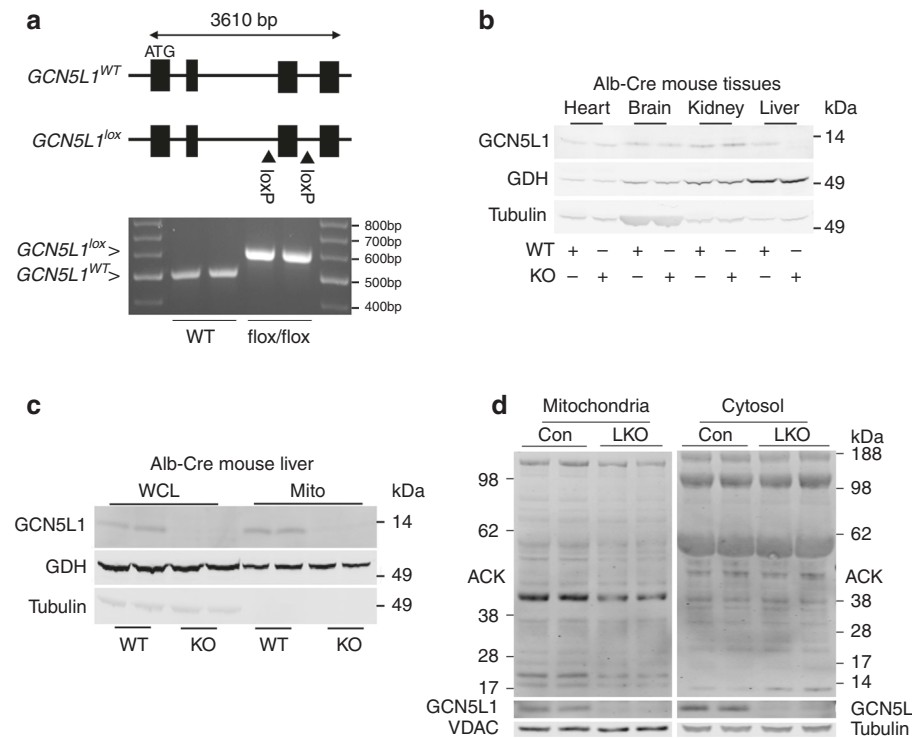

**Fig. 1** GCN5L1 liver knockout attenuated liver mitochondrial protein acetylation. **a** Schematic of the GCN5L1 gene showing exon 3 flanked by two loxP sites and the subsequent recombination with mice harboring the albumin-Cre-Recombinase to facilitate excision of exon 3. **b** Immunoblot showing GCN5L1 expression in heart, brain, kidney, and liver of wild type (WT) and GCN5L1 LKO mouse tissue lysates. **c** Immunoblot showing GCN5L1 levels in whole liver and isolated mitochondrial lysates of WT and LKO mice. **d** Immunoblot analysis of mitochondrial protein acetylation and cytosolic protein acetylation in GCN5L1 LKO mice and control (flox/flox) littermates. 30 μg of isolated mitochondrial protein and 60 μg of cytosolic protein were loaded for immunoblot analysis (n = 4 independent experiments)

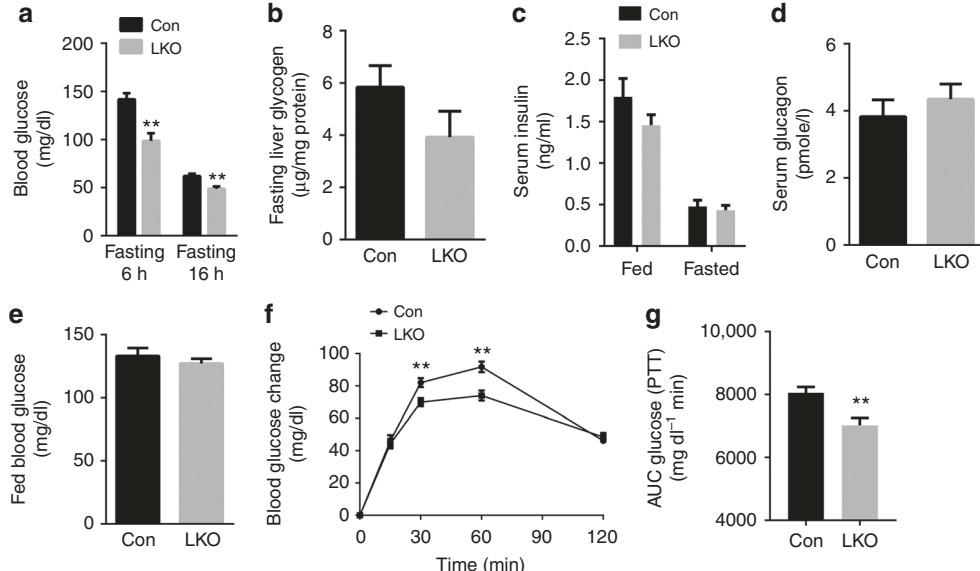

**Fig. 2** GCN5L1 LKO mice displayed decreased fasting blood glucose levels. 8–10 weeks male mice fed with normal chow were studied. **a** Blood glucose levels were measured under fasting conditions. Mice were fasted for 6 or 16 h ($n = 7$). **b**–**d** Liver glycogen content after 6 h fasting ($n = 5$) (**b**), fed and overnight fasted serum insulin ($n = 7$) (**c**) and serum glucagon levels after 6 h of fasting ($n = 5$) (**d**). **e** Blood glucose levels were measured under fed conditions ($n = 6$). **f**, **g** Pyruvate tolerance test (PTT) and area under the curve of blood glucose levels ($n = 18$). Values are expressed as mean ± s.e.m. **P < 0.01 versus respective control groups by the Student's $t$-test

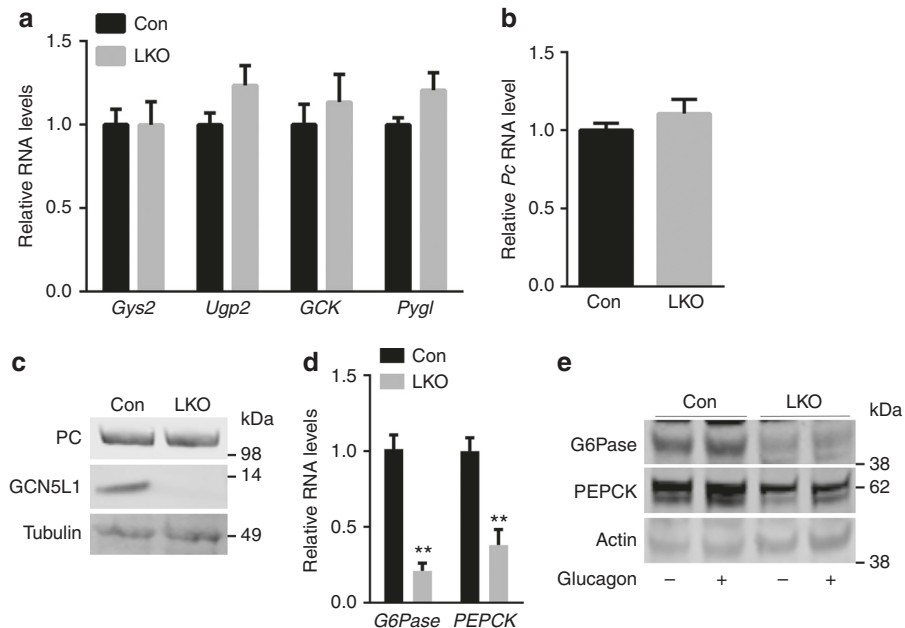

**Fig. 3** GCN5L1 LKO downregulated cytosolic gluconeogenic enzymes. **a** The transcript levels of the rate-controlling enzymes of glycogenesis and glycogenolysis in control and LKO primary hepatocytes ($n = 5$ paired sets of primary hepatocytes). **b**, **c** Mitochondrial gluconeogenic enzyme pyruvate carboxylase(PC) RNA (**b**) and protein (**c**) levels in primary hepatocytes ($n = 5$ paired sets of primary hepatocytes). **d** RNA levels of cytosolic rate controlling enzymes G6Pase and PEPCK in primary hepatocytes ($n = 6$ paired sets of primary hepatocytes). **e** Primary hepatocytes were isolated and stimulated with or without glucagon (100 nM) for 6 h followed by the assessment of the levels of G6Pase and PEPCK by immunoblot analysis. Values are expressed as mean ± s.e.m. **P < 0.01 versus respective control groups by Student's $t$-test. Glycogen synthase 2 (Gys2), UDP-glucose pyrophosphorylase 2 (Ugp2), Glucokinse (GCK) and Glycogen phosphorylase L (Pygl)

Here, GCN5L1 depletion links to increased FoxO1 phosphorylation, ubiquitylation and proteasome-dependent degradation. This FoxO1 phenotype is dependent on excess mitochondrial ROS-dependent ERK activation in the absence of GCN5L1 and is partially reversed by inhibition of ERK signaling. Additionally, the reconstitution of mitochondrial-targeted GCN5L1 blunts mitochondrial ROS, reduces ERK activation, restores FoxO1 levels, upregulates PEPCK and G6Pase transcript levels and restores gluconeogenesis. These findings support that mitochondrial GCN5L1 levels modulate the post-translational control of

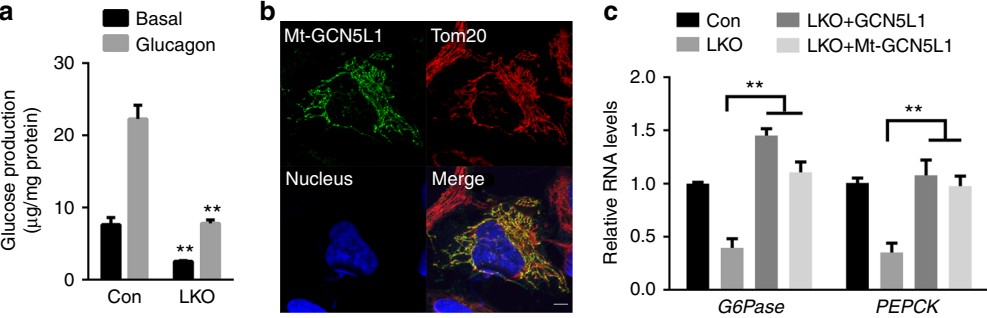

**Fig. 4** GCN5L1 LKO impaired gluconeogenesis. **a** Glucose production was measured in the culture medium of primary hepatocytes isolated from GCN5L1 LKO and littermate controls at 5 h after treatment with or without 100 nM glucagon. Glucose production was normalized to protein content ($n = 3$ independent experiments). **b** HeLa cells were transiently transfected with Mt-GCN5L1 plasmid, followed by immunofluorescence staining and confocal analysis. Tom20 serves as mitochondrial marker. Scale bar, 5 μm. **c** Primary hepatocytes were isolated and transfected with adenoviral expression of vector control, GCN5L1, or Mt-GCN5L1. RNA was analyzed at 36 h after transfection ($n = 3$ independent experiments). Values are expressed as mean ± s.e.m. **P < 0.01 versus respective groups by Student's t-test

FoxO1, and this regulation of gluconeogenesis occurs, in part, via mitochondrial ROS control of ERK signaling. The direct link between hepatic GCN5L1 depletion and mitochondrial ROS generation remains to be elucidated.

## Results
**Basal phenotyping of liver-specific GCN5L1 knockout mice.**
Using our previously described GCN5L1 Lox-P mice[4], liver-specific knockout (LKO) mice were generated following crossing with albumin Cre-recombinase mice. The genetic strategy to disrupt GCN5L1 is shown (Fig. 1a). GCN5L1 LKO mice are born in a normal Mendelian distribution and are viable with no overt signs of illness up to 10 months of age. Immunoblot analysis of whole tissue from the heart, brain, kidney, and liver, and samples from the liver whole cell and mitochondrial fractions show restricted ablation of GCN5L1 in liver tissue and in liver whole cell lysate and mitochondria (Figs. 1b, c). In parallel with cell culture studies our findings from our laboratory and others[1, 3, 4, 7], the livers from LKO mice show a modest reduction in basal mitochondrial protein acetylation with no change in cytosolic acetylation levels compared to littermate controls (Fig. 1d).

The body and liver weights (Supplementary Fig. 1a, b) are no different comparing the control (flox/flox) and LKO mice. In the fed state liver triglyceride levels are similar (Supplementary Fig. 1c), although there is a modest, albeit significant increase in liver cholesterol levels (Supplementary Fig. 1d) and reduced liver glycogen levels (Supplementary Fig. 1e) in LKO mice.

Under fasting conditions, the LKO mice show a significant reduction in blood glucose levels (Fig. 2a) with dissipation in the difference in liver glycogen levels (Fig. 2b) and show similar levels of serum insulin (Fig. 2c) and glucagon (Fig. 2d) compared to littermate controls. Interestingly, in wildtype mice livers the level of GCN5L1 is significantly induced at the 6 h fasting time point, which correlates with maximal relative blunting of glucose levels in LKO mice (Supplementary Fig. 2a). In parallel, there is no difference in fed blood glucose (Fig. 2e) or in glucose levels in response to glucose and insulin tolerance testing between the strains (Supplementary Fig. 2b, c). In contrast to insulin and glucose tolerance tests, but in keeping with the difference in fasting glucose levels, the LKO mice show a modest, but significant reduction in glucose production in response to a pyruvate challenge (Figs. 2f, g). The exploration of hepatic insulin signaling in vivo shows no discernable differences in insulin-dependent phosphorylation of the insulin receptor and of Akt in LKO mice (Supplementary Fig. 2d).

**GCN5L1 LKO mice have impaired hepatocyte gluconeogenesis.**
The baseline phenotyping data suggest that GCN5L1 LKO mice may have impaired fasting hepatic glucose production. To explore this pathway subsequent studies were performed in primary hepatocytes. We firstly quantified the transcript levels of genes encoding the rate controlling steps of glycogenesis and glycogenolysis and found no differences in gene expression comparing the control and LKO hepatocytes (Fig. 3a). At the same time the expression of the mitochondrial enriched gluconeogenesis enzyme, pyruvate carboxylase (PC), is also not altered at the transcript or protein level by the disruption in GCN5L1 (Figs. 3b, c). In contrast, the genes encoding the pivotal cytosolic gluconeogenic enzymes, phosphoenolpyruvate carboxykinase (PEPCK) and glucose-6-phosphatase (G6Pase) are significantly downregulated at the transcript (Fig. 3d) and at the steady-state protein level at baseline and in response to glucagon stimulation (Fig. 3e). Interestingly, this regulatory effect is less prominent in the intact LKO liver, where G6Pase transcript levels are significantly downregulated, but PEPCK mRNA levels only trended lower (Supplementary Fig. 3a).

Primary hepatocytes were then maintained in glycogen depletion media overnight and glucose production from pyruvate was assayed at baseline and in response to glucagon. Under basal conditions the production of glucose is profoundly blunted in the LKO hepatocytes (Fig. 4a). To confirm whether the disruption of gluconeogenesis was dependent on GCN5L1, reconstitution experiments were performed in the LKO primary hepatocytes. Here, the reintroduction of WT and mitochondrial targeted (Mt) GCN5L1 (Fig. 4b) restore transcript levels of PEPCK and G6Pase (Fig. 4c).

**GCN5L1 knockout disrupts gluconeogenic regulatory signals.**
The nuclear regulatory control of PEPCK and G6Pase transcription are well defined and appear to segregate temporally into acute and more chronic engagement of regulatory proteins to orchestrate gluconeogenesis[8–11]. The glucagon-dependent acute fasting induction of gluconeogenesis is dependent on CREB and CRTC transcript activation[12]. In parallel with the similar relative rate of increase glucose production in response to glucagon, we find no difference in glucagon-induced CREB phosphorylation (Fig. 5a) or in CRTC2-CREB activation of a CRE-luciferase construct (Fig. 5b) in control and LKO hepatocytes. Interestingly, the relative ratio of glucagon-induced glucose production is similar in WT and LKO hepatocytes, suggesting that the GCN5L1 effect may be glucagon independent (Fig. 5c).

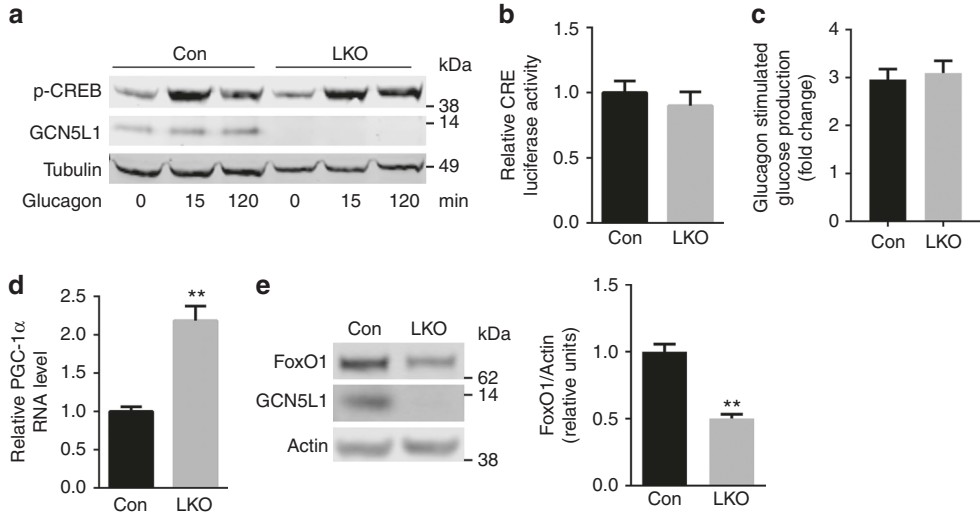

**Fig. 5** Gluconeogenic regulatory signals were disrupted in the absence of GCN5L1. **a** Primary hepatocytes were isolated from LKO mice and littermate controls and treated with glucagon (100 nM) at times indicated. Cell lysates were subjected to immunoblotting with the indicated antibodies. **b** Primary hepatocytes were transfected for 24 h with Ad-CRE luciferase and Ad-β-galactosidase (as infection control) and followed by glucagon incubation for 5 h. Luciferase activity corrected for transfection efficiency was assayed in cell lysates ($n = 3$ independent experiments). **c** Glucagon stimulated glucose production was calculated by normalizing glucose production after glucagon stimulation to basal condition of each group from Fig. 4a. **d** PGC-1α RNA level ($n = 5$ paired sets of primary hepatocytes) in primary hepatocytes. **e** Cell lysates from primary hepatocytes were used for immunoblot analysis to assess the levels of FoxO1 ($n = 5$ paired sets of primary hepatocytes). Values are expressed as mean ± s.e.m.**$P < 0.01$ versus respective control groups by Student's t-test

Although PGC-1α has been shown to upregulate PEPCK and G6Pase we find that the PGC-1α transcripts are higher in LKO hepatocytes (Fig. 5d) and liver (Supplementary Fig. 3b) despite lower expression of these gluconeogenic enzyme transcripts. The induction of PGC-1α has previously been found in primary GCN5L1 KO MEF cells[4], although here, this induction does not correlate with the expression of PEPCK and G6Pase. In contrast, the steady-state levels of FoxO1, a known transactivator of gluconeogenesis in response to fasting[12], is significantly lower in GCN5L1 LKO hepatocytes (Fig. 5e), and the fasting levels of FoxO1 are similarly lower in GCN5L1 LKO livers (Supplementary Fig. 3c).

**Mitochondrial enriched GCN5L1 modulates FoxO1 stability.** Given that FoxO1 is an essential transcriptional regulator of gluconeogenesis we focused our further studies on the role of GCN5L1 on the regulation of FoxO1. To explore the level/s of regulation of FoxO1 we first assayed FoxO1 transcript levels and find that there is no difference in expression between control and LKO hepatocytes (Fig. 6a). To assess whether the absence of GCN5L1 is responsible for the reduced FoxO1 steady state levels, we assessed whether reintroduction of GCN5L1 into LKO hepatocytes restored FoxO1 levels. Here, the restoration of both WT and Mt-GCN5L1 in the LKO hepatocytes restore FoxO1 protein levels (Fig. 6b). We then assayed FoxO1 protein turnover by exposing hepatocytes to cycloheximide. Here, the rate of FoxO1 degradation is significantly higher in LKO hepatocytes (Fig. 6c). To identify the degradation pathway/s operational following GCN5L1 depletion, primary hepatocytes were exposed to endosome/lysosomal inhibitors (bafilomycin A and chloroquine) and proteasome inhibitors (lactacystin and MG132). Only the proteasome inhibitors prevent the degradation of FoxO1 (Fig. 6d). Given these findings we assessed the extent of FoxO1 ubiquitylation in the presence and absence of MG132. In vehicle treated hepatocytes, FoxO1 ubiquitylation is only modestly higher in the LKO cells. However, in the presence of MG132, this effect is markedly accentuated with robust ubiquitylation of FoxO1. Again, reconstitution with Mt-GCN5L1 in the LKO cells reduces FoxO1

ubiquitylation under these conditions (Fig. 6e). The functional role of diminished FoxO1 in GCN5L1 LKO blunting of gluconeogenesis is confirmed following FoxO1 overexpression in GCN5L1 LKO hepatocytes by the restoration of PEPCK and G6Pase transcript levels (Fig. 6f) and the partial restoration of glucose production (Fig. 6g). However, given the central role of FoxO1 in regulating gluconeogenesis, these data cannot exclude additional mechanisms whereby FoxO1 overexpression rescues gluconeogenesis in the GCN5L1 null hepatocytes.

**GCN5L1 regulates ERK control of FoxO1 stability and gluconeogenesis.** Given that protein phosphorylation may precede ubiquitylation in the promotion of protein degradation[16], we compared the phosphoproteome of Flag-tagged FoxO1 overexpressed in WT versus LKO primary hepatocytes. We find that multiple serine/threonine residues on FoxO1 are excessively phosphorylated in the absence of GCN5L1 (Supplementary Data 1). The most highly differentially phosphorylated residues are found to be validated targets of ERK and p38 MAP kinase[17] (Supplementary Fig. 4a). We then explored whether p38 MAPK and MEK/ERK inhibitors modulated FoxO1 levels and gluconeogenesis in the LKO genetic background. The p38 inhibitor SB203580 does not restore FoxO1 protein levels nor the gluconeogenic gene transcript or glucose production in GCN5L1 KO cells (Figs. 7a–c). In contrast, the ERK inhibitor U0126 has robust effects in restoring FoxO1, its canonical gluconeogenesis transcripts and glucose production levels in the LKO hepatocytes (Figs. 7a–c). A second ERK inhibitor has a less robust, albeit significant effect on restoring gluconeogenic transcripts and glucose production in the LKO hepatocytes (Supplementary Fig. 4b, c). The role of ERK activation in blunting gluconeogenesis is supported by higher levels of ERK phosphorylation in primary LKO hepatocytes (Fig. 7d) and in the liver (Supplementary Fig. 4d) and by the finding that overexpression of WT (G) and Mt GCN5L1(MTG), reverse ERK phosphorylation (Fig. 7e). In contrast, endoplasmic reticulum targeted GCN5L1 (ERG) does not modulate the levels of ERK phosphorylation (Fig. 7e).

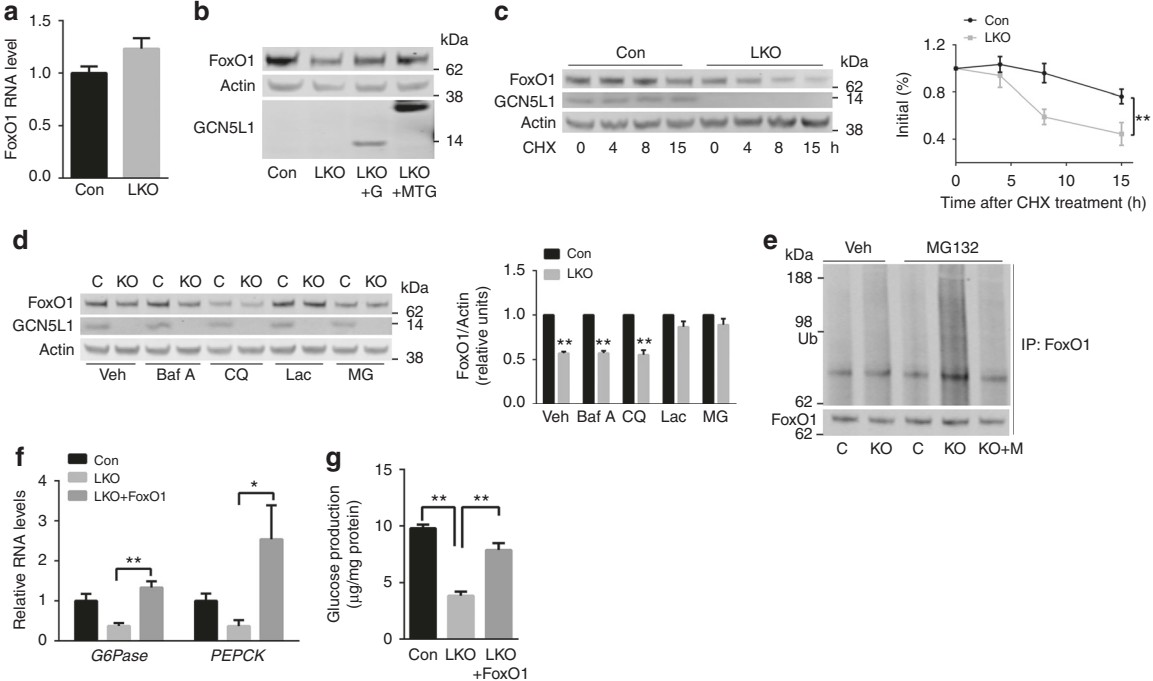

**Fig. 6** GCN5L1 regulates gluconeogenesis through FoxO1 stability. **a** FoxO1 RNA expression levels in primary hepatocytes ($n = 4$ paired sets of primary hepatocytes). **b** Cell lysates from primary hepatocytes transfected with Ad-control, Ad-GCN5L1(G) or Ad-Mt-GCN5L1 (MTG). A representative immunoblot shows the subsequent analysis of FoxO1 levels ($n = 3$ independent experiments). **c** Representative immunoblot of cycloheximide (CHX, 50 mg ml$^{-1}$)-treated primary hepatocytes at times indicated. The quantification of FoxO1 was normalized to β-Actin, relative to time 0 as 100% for each group ($n = 3$ independent experiments, **$P < 0.01$, evaluated by two-way ANOVA). **d** Primary hepatocytes were isolated and incubated with endosome/lysosomal inhibitors bafilomycin A (Baf A) or chloroquine (CQ), or proteasome inhibitors lactacystin (Lac) or MG132 (MG) for 18 h. Cell lysates were used in immunoblot analysis for FoxO1 levels. Quantification of FoxO1 was normalized to β-Actin, relative to control for each treatment ($n = 3$ independent experiments, **$P < 0.01$, evaluated by Student's t-test). **e** Western blot analysis of ubiquitylation in Mt-GCN5L1(M) transfected primary hepatocytes following immunoprecipitation with anti-FoxO1, with or without MG132 ($n = 3$ independent experiments). **f, g** Primary hepatocytes were transfected with FoxO1. RNA levels of G6Pase and PEPCK (**f**) and glucose production (**g**) were analyzed at 36 h after transfection ($n = 3$ independent experiments, *$P < 0.05$; **$P < 0.01$, evaluated by Student's t-test)

**GCN5L1 knockout increases ROS to suppress gluconeogenesis.** Given that ERK can be activated by hydrogen peroxide ($H_2O_2$)[18], we then explored if the GCN5L1 effects on mitochondria may modulate mitochondrial ROS levels. We show that ROS levels are significantly higher in LKO hepatocytes by measuring CM-H2DCFDA using flow cytometry and in LKO hepatocyte isolated mitochondria by assaying $H_2O_2$ using Amplex Red (Figs. 8a, b). This elevated ROS in the GCN5L1 KO hepatocyte mitochondria is completely abrogated by either diphenylene iodonium (DPI), an inhibitor of flavin-containing cofactors which has been found to inhibit mitochondrial ROS production[19], and with the superoxide dismutase mimetic MitoTEMPO (Figs. 8a, b). Both these antioxidant compounds similarly blunt excessive ERK phosphorylation (Fig. 8c). MitoTEMPO also partially and significantly restores G6Pase and PEPCK transcript levels and glucose production in the LKO hepatocytes (Figs. 8d, e). In parallel, the restoration of GCN5L1 in LKO hepatocytes concordantly blunts excess ROS levels (Figs. 8f, g). Finally, and consistent with the effects of GCN5L1 reconstitution on gluconeogenic gene transcript levels, FoxO1 stability and ERK phosphorylation, the rescue of LKO hepatocytes with WT or Mt-GCN5L1 similarly restores glucose production (Fig. 8h).

## Discussion

The concept that changes in mitochondrial function or metabolism can initiate retrograde signaling from mitochondria to the nucleus has come to the fore[6, 20]. In parallel, the genetic depletion of GCN5L1 has previously been found to initiate mitonuclear retrograde signaling via the activation of PGC-1α and TFEB[4]. In this study, we extended the understanding of the role of mitochondrial GCN5L1, where we show that GCN5L1 levels, via the modulation of mitochondrial ROS results in ERK mediated post-translational control of the canonical gluconeogenic transcription factor, FoxO1. Interestingly, this ERK-dependent control of FoxO1 appears to be an insulin signaling-independent pathway in the control of gluconeogenesis.

The role of acetyl-CoA and protein acetylation on the control of gluconeogenesis was initially implicated in a yeast deacetylase microarray studies[21] with subsequent validation where the rate-controlling gluconeogenic enzyme PEPCK-C was activated by acetylation in yeast and in human HepG2 cells[15]. More recently, in vivo hepatic exposure to excessive acetyl-CoA induced gluconeogenesis, in part, via the activation of the mitochondrial gluconeogenic enzyme pyruvate carboxylase[14]. In contrast, deacetylation of nuclear regulatory proteins driving gluconeogenesis including FoxO1 and PGC-1α have been found to transactivate gluconeogenic encoding genes and increase hepatic glucose production[22, 23] and the nuclear GCN5 acetyltransferase complex inhibited gluconeogenesis via acetylation of PGC-1α[9]. Together, these data support that acetyl protein modifications, within distinct subcellular compartments, and possibly under different nutrient conditions[12], may have different effects on the same metabolic pathway.

Given that we found a restricted reduction in mitochondrial protein acetylation in parallel with reduced glucose production in GCN5L1 LKO hepatocytes, we explored this pathway to further

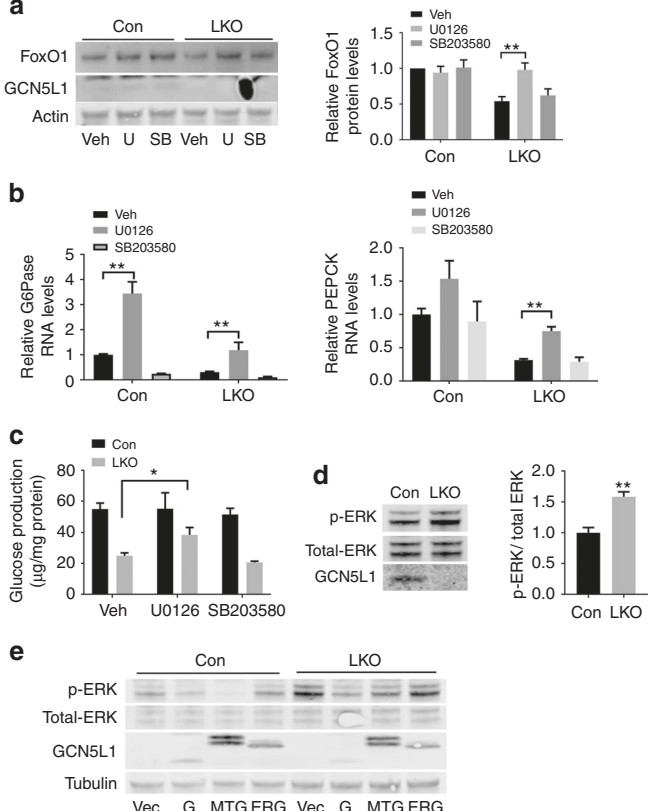

**Fig. 7 Mitochondrial GCN5L1 controls FoxO1 stability via controlling ERK activation**. **a–c** Primary hepatocytes were isolated and incubated with U0126 (U) or SB203580 (SB) for 8 h. Cell lysates were analyzed by immunoblotting. ($n = 3$ independent experiments). **a** Quantification of FoxO1 was normalized to β-Actin, relative to vehicle treated control. **b** RNA levels of G6Pase and PEPCK were analyzed by real-time PCR ($n = 3$ independent experiments). **c** Glucose production was assayed by secretion into the media from these cells ($n = 3$ independent experiments). **d** Immunoblot analysis to assay the activation of ERK in primary hepatocytes. Quantification of p-ERK was normalized to total-ERK, relative to control group ($n = 5$ paired sets of primary hepatocytes). **e** Primary hepatocytes were isolated and infected with adenovirus expressing GCN5L1(G), Mt-GCN5L1(MTG) or ER-GCN5L1(ERG) for 24 h. Cell lysates were analyzed by immunoblotting of p-ERK. Values are expressed as mean ± s.e.m. *$P < 0.05$; **$P < 0.01$ by Student's $t$-test

understanding of the role of mitochondrial GCN5L1 in the control of gluconeogenesis. The initial analyses of rate-controlling enzymes in gluconeogenesis and glycogenolysis show that the transcript and protein levels of the cytosolic enzymes controlling gluconeogenesis including PEPCK and G6Pase are substantially lower in GCN5L1 LKO hepatocytes without changes in the levels of the mitochondrial enzyme pyruvate carboxylase. Given this reduction in transcript levels, we then explored the known pathways involved in the transcriptional control of gluconeogenesis. It should be noted that the transcriptional programs controlling gluconeogenesis differ between early and more prolonged fasting[12]. During early fasting, the CREB mediated transcriptional program is operational and this switches to a FoxO1 driven program as fasting becomes more prolonged[12]. In the absence of GCN5L1, we find that the FoxO1 pathway is altered, given the reduced steady-state levels of this transcription factor. At the same time GCN5L1 is significantly induced in wildtype mice within six hours of fasting. Whether GCN5L1 is an important regulatory component in the transition from acute to longer term fasting could be important. However, this temporal aspect of this regulation was not explored further in this study. Nevertheless, the decrease in FoxO1 levels appears to be regulated by its ubiquitylation and degradation through the ubiquitin-proteasome pathway. As studies have shown post-translational modification cross-talk linking protein phosphorylation and ubiquitylation with protein turnover[24, 25] we then explored the GCN5L1-level dependent phosphorylation of

FoxO1. Mass spectroscopy identified multiple canonical ERK and p38 MAPK kinase serine residues with markedly higher phosphorylation in the LKO hepatocytes. Here, the pharmacologic inhibition of ERK, but not p38 MAPK increased FoxO1 levels, restored gluconeogenic gene transcript levels and glucose production. In parallel, reconstitution with FoxO1 and mitochondrial directed GCN5L1 similarly restores gluconeogenesis and glucose production. Putting these data together, it appears that the depletion of GCN5L1 promotes ERK activation with the subsequent cascade of events resulting in reduced glucose production in hepatocytes.

The signaling-transduction pathways modulating gluconeogenesis have been relatively well explored. The major mediators include cyclic-AMP-mediated transactivation and insulin-induced phosphatidylinositol 3-kinase (PI3K)/Akt signaling mediated inhibition of gluconeogenic genes and gluconeogenesis[26, 27]. Even though ERK is downstream of insulin signaling, its role has predominantly been explored in the induction of protein synthesis[27]. Recent data does, however, implicate ERK in the phosphorylation of, and cytosolic retention of FoxO1 to downregulate gluconeogenesis[28]. In addition, a study exploring the role of TGF-β1/Smad3 in gluconeogenesis, found that the suppression of hepatic gluconeogenesis by Smad3 KO was associated with a robust induction of ERK1/2 phosphorylation[29]. Here, we further extend this function of ERK by showing that its activation phosphorylates and promotes the degradation of FoxO1. In addition to the reduction in total FoxO1 levels in

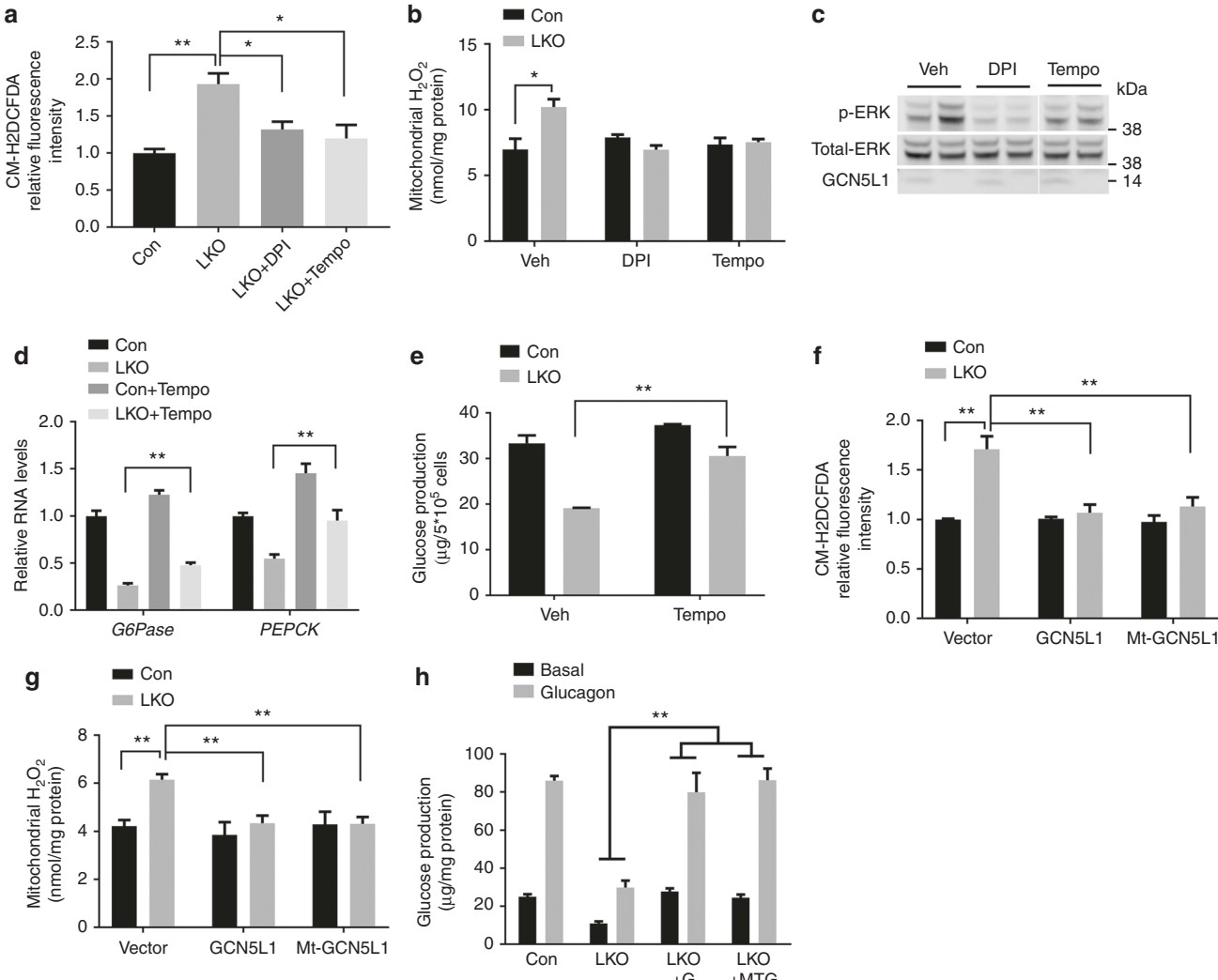

**Fig. 8** GCN5L1 knockout resulted in excessive ROS signaling in the suppression of gluconeogenesis. **a** Cellular ROS levels were measured by FACS. Primary hepatocytes were incubated with vehicle (Veh), Diphenyleneiodonium (DPI) or MitoTEMPO (Tempo) for 30 min before incubated with CM-H2DCFDA. Mean fluorescence intensity was normalized to Con group with Veh treated ($n = 3$ independent experiments). **b** Mitochondrial hydrogen peroxide levels were measured using isolated mitochondria from DPI or MitoTEMPO treated primary hepatocytes ($n = 3$ independent experiments). **c** Primary hepatocytes were treated with DPI or MitoTEMPO for 30 min and cell lysates were analyzed by immunoblotting of p-ERK ($n = 3$ independent experiments). **d**, **e** Primary hepatocytes were incubated with Veh or MitoTEMPO overnight and followed by another 4 h incubation in glucose production medium for RNA analysis (**d**) and glucose analysis (**e**) ($n = 3$ independent experiments). **f** Cellular ROS levels were measured in hepatocytes at 36 h after adenovirus infection. Mean fluorescence intensity was normalized to Con group with vector adenovirus infected ($n = 3$ independent experiments). **g** Mitochondrial hydrogen peroxide levels were measured using isolated mitochondria from adenovirus infected primary hepatocytes ($n = 3$ independent experiments). **h** Primary hepatocytes were isolated and transfected with adenoviral expression of control, GCN5L1 (G) or Mt-GCN5L1 (MTG), glucose production was analyzed at 36 h after transfection ($n = 3$ of independent experiments). Values are expressed as mean ± s.e.m. *$P < 0.05$, **$P < 0.01$ by Student's $t$-test

GCN5L1 LKO, we also find that its nuclear sequestration was similarly diminished (Supplementary Fig. 5). Interestingly, and in contrast with what would be expected with blunted gluconeogenesis, there was no increase in phosphorylation of the insulin receptor or Akt in LKO mice livers in response to insulin. This latter finding suggests that GCN5L1-depletion-mediated downregulation of gluconeogenesis is independent of insulin signaling.

In pilot experiments, we employed siRNA knockdown studies to explore the role of known FoxO1 degrading proteins including XBP-1, COP1, SKP-2 and MDM-2[30-33]. The genetic depletion of these candidates did not restore glucose production or transcripts of gluconeogenic enzymes in the GCN5L1 LKO hepatocytes (Supplementary Fig. 6). Going forward an unbiased approach will be required to dissect out the molecular mediator directly orchestrating FoxO1 ubiquitylation. Interestingly, in a prior study

the restricted hepatic genetic depletion of FoxO1 only reduced glucose production by approximately 30%[34] suggesting that there is significant redundancy in the control of this essential hepatic glucose production pathway. Mitochondrial metabolism itself may be one such component in the control of glucose production, given the inter-mitochondrial biochemical role in gluconeogenesis[35] and even the role of mitochondrial function in the regulation of insulin signaling[36]. Although we did not find disruption in mitochondria respiratory function or in mitochondrial genomic or protein context in the LKO hepatocytes, we show that the generation of mitochondrial ROS plays an important role in this signaling. We show that mitochondrial ROS levels are elevated in the absence of GCN5L1 and that the reduction in ROS levels blunts ERK activation in GCN5L1 LKO hepatocytes. Furthermore, reconstitution with WT and Mt-

GCN5L1 in LKO hepatocytes similarly blunts ROS levels and restores the gluconeogenesis pathway and glucose production. Together these data suggest that overall mitochondrial disruption per se did not play a major role in this regulation of gluconeogenesis, although retrograde ROS-mediated signaling from mitochondria is operational, following the genetic ablation of GCN5L1. Whether other mitochondrial acetylation dependent effects contribute to the phenotype of reduced gluconeogenesis require additional exploration. At the same time, it should be noted that although they have significant effects, the rescue by FoxO1 overexpression and ERK and ROS inhibition did not fully rescue blunting of gluconeogenesis in GCN5L1 knockout hepatocytes. These data suggest that additional GCN5L1 regulatory pathways may be operational in gluconeogenesis control. At the same time, the more robust effects in primary cells versus in-vivo could suggest a degree of compensation for the loss of GCN5L1 in hepatic gluconeogenesis.

Overall, the concept that mitochondrial acetylation levels modulate the transcriptional control of a metabolic pathway is compatible with the finding of increased glycolysis, the reciprocal program to gluconeogenesis, in SIRT3 knockout mice through retrograde signaling mediated stabilization of HIF1α[37] and the prior demonstration that GCN5L1 knockout MEF cells evoked the upregulation of both PGC-1α and TFEB[4]. Furthermore, the concept that the alteration in mitochondrial protein acetylation can trigger retrograde cytosolic and nuclear reprogramming is also increasingly being reported[38, 39].

The ongoing investigation into the multiple levels of regulation of gluconeogenesis reflects the need to understand this pathway to enable the identification of potential targetable regulatory nodes that could be exploited for novel pharmacologic intervention to ameliorate excessive gluconeogenesis in Type 2 diabetes mellitus. The findings in this study identify that the deacetylation of mitochondrial proteins in the absence of GCN5L1 played a regulatory role in controlling gluconeogenesis via ROS-mediated retrograde control of ERK/FoxO1-dependent signaling and transcription. Whether this GCN5L1 regulatory pathway would be amenable to modulation to blunt gluconeogenesis requires additional characterization.

## Methods

**Animal studies.** All animal protocols were in accordance with Institutional Guidelines and approved by the Institutional Animal Care and Use Committee of the National Heart, Lung and Blood Institute of the National Institutes of Health, USA. The mice were maintained on a 12-hour light/dark cycle and housed three to five mice per cage with free access to water and normal chow diet. GCN5L1 liver knockout mice were generated by crossing GCN5L1 flox/flox mice with albumin-Cre flox mice. All mice were generated in the C56BL/6 background. For insulin tolerance testing, 10–11-week male mice fed with normal chow were fasted 4 h, followed by intraperitoneal injection of insulin (0.75 IU/kg body weight), blood glucose was measured by tail bleeding at 0, 30, 60, 90, and 120 min after injection. For pyruvate tolerance and glucose tolerance testing, 9–12-week male mice were fasted overnight, followed by i.p. injection of sodium pyruvate (2 g kg$^{-1}$ body weight) or glucose (2 g kg$^{-1}$ body weight). Mice were not randomized as there we no interventions and WT mice were directly compared to LKO mice. The studies were not blinded. Timed blood draws for glucose measurement was done through tail bleeding. In vivo insulin signaling pathway analysis was performed in 10–11-week male mice following an overnight fast. Mice were anesthetized, and insulin (2 IU kg$^{-1}$) or PBS was injected through the portal vein. Three minutes later, livers were removed, frozen in liquid nitrogen, and stored at −80 °C. For immunoblot analysis, tissues were homogenized in RIPA buffer, 80 μg total protein was used for each sample.

**Cell culture studies.** Primary hepatocytes were isolated from normal chow fed mice at age of 8–12 weeks. Mice was anesthetized with avertin, and their livers was perfused with Krebs Ringer buffer with glucose (135 mM NaCl, 5 mM KCl, 1 mM MgSO$_4$, 0.4 mM K$_2$HPO$_4$, 20 mM HEPES, and 20 mM glucose, pH 7.4) and EGTA (100 mM) for 3 min, followed by continuous perfusion with the same buffer containing CaCl$_2$ (1.4 mM) and collagenase (7000 IU per mouse, type I, Worthington) for 8 min. The liver was then removed and the isolate hepatocytes were filtrated through a 100-μm cell strainer and purified by Percoll (Sigma). Hepatocytes were plated (6 × 10$^5$ cells per well in six-well plate) into collagen (Sigma) coated plates and cultured in DMEM medium containing 10% FBS and 1% p/s.

For putative degradation pathway studies, Bafilomycin A1 (10 nM), Cholorquine (25 μM), Lactacystin (5 μM) or MG132 (5 μM) was added to the culture medium for 18 h.

To inhibit ERK and p38 MAP kinase, 10 μM U0126 or SB203580 or 50 μM PD98059 were incubated with primary hepatocytes for 8 h in glucose free medium. The medium was then assayed for glucose production and cells were collected for RNA and protein analysis.

To inhibit ROS generation, 0.5 μM DPI or 5 μM MitoTEMPO were incubated with hepatocytes for 30 min in glucose free medium for FACS and western blot analysis of ERK. Mito-TEMPO was incubated 15 h with Primary hepatocytes were preincubated with Mito TEMPO for 15 h and sustained for an additional 4–6 h in fresh glucose-free medium for subsequence measurement of glucose production and RNA analysis.

HeLa cells (ATCC) were used to confirm the localization of the mitochondrial-targeted GCN5L1.

**Glucose production assay.** Glucose production from primary hepatocytes was measured using a colorimetric glucose oxidase assay (Sigma). Briefly, primary hepatocytes were isolated and cultured in DMEM medium containing 10% FBS for 24 h, and then were cultured in glucose production medium (glucose-free DMEM, pH 7.4, containing 20 mM sodium lactate and 2 mM sodium pyruvate, without phenol red) overnight to deplete glycogen, followed by incubated for another 5 h at 37 °C, 5% CO$_2$, in glucose production buffer with or without 100 nM glucagon. The medium was collected for glucose measurement. The glucose assays were conducted in triplicate.

**Plasmid constructs and adenovirus production.** A Flag-tagged FoxO1 expression construct was purchased from Addgene. Mitochondria-GCN5L1 was generated by fusing the GCN5L1 encoding sequence with the COX 8 mitochondrial targeting sequence (36 amino acid at N terminal) and 3×-flag sequence. ER-targeted GCN5L1 was generated by fusing GCN5L1 encoding sequence with the N terminal 17 amino acids of calreticulin at the N terminal and KDEL and 3×-flag at the C terminal. GCN5L1 DNA, mitochondria-GCN5L1 and ER-GCN5L1 were cloned into pShuttle-CMV vector (Agilent), or an empty vector as a negative control. Adenoviruses were produced using Adeasy Adenoviral System (Agilent). After amplification with Ad-293 packaging cell line, virus was purified using ceasium chloride gradient ultracentrifugation and dialyzed into PBS with desalting columns (GE). Adenovirus of flag tagged FoxO1 was purchased from Vector Biolabs. Primary hepatocytes were infected with adenovirus overexpressing either empty vector (control), GCN5L1, Mt-GCN5L1, ER-GCN5L1 or Flag-FoxO1 at a dose of 20 p.f.u per cell. Thirty-six hours after transduction, cells were used for experiments.

**Chemicals and antibodies.** The antibodies were purchased as follows: FoxO1 (2880), VDAC(4661), Tubulin(2146), Tom20(42406, dilution 1:100 for IF), p-ERK (9101), total ERK(9102), p-IRβ (Y1146) (3021), p-AKT (S473) (9271), AKT(9272); ACK(9441), Ubiquitin(3936) were from Cell Signaling Technology, and used dilution 1:1000 for western blot; G6Pase(sc-32840), Actin(sc-1615) and PEPCK(sc-74825) were from Santa Cruz, and used dilution 1:1000 for western blot; total IR(07-724) was purchased from Millipore, and used dilution 1:1000 for western blot. The fluorescent secondary antibodies were as follows: Alexa Fluor 488 donkey anti-mouse IgG and Alexa Fluor 488 donkey anti-rabbit IgG were from Life Technologies. U0126, PD98059 and SB203580, Diphenylene iodonium (DPI) and MitoTEMPO were purchased from Sigma.

**Mitochondria isolation.** Mitochondria were isolated from primary hepatocytes or mouse livers. Cells or liver tissues were homogenized in mitochondrial isolation buffer (225 mM mannitol, 75 mM sucrose, 0.5% BSA, 0.5 mM EGTA, and 30 mM Tris–HCl pH 7.4), and followed by two-step centrifuge to collect the mitochondria.

**Ubiquitylation analysis.** Primary hepatocytes were isolated from GCN5L1 LKO mouse and littermate controls. The following day, the cells were incubated in glucose production medium with or without 10 μM MG132 for 4 h, and then cells were lysed with lysis buffer (20 mM Tris-HCl, pH 7.5, 137 mM NaCl, 5 mM EDTA, 0.5% NP-40, and protease inhibitor cocktail) for 30 min at 4 °C. The homogenates were centrifuged for 15 min at 12,000 r.p.m. at 4 °C. The supernatant was incubated with indicated antibodies overnight at 4 °C. Protein A/G plus agarose beads (Santa Cruz) were added at 4 °C for another 4 h. The immunoprecipitation beads were washed with wash buffer (20 mM Tris-HCl, pH 7.5, 150 mM NaCl, 5 mM EDTA, and 1% Triton X-100) for three times, followed by western blotting analysis.

**Mass spectrometry of FoxO1 modification.** Primary hepatocytes from WT and GCN5L1 LKO mice were transduced with adeno-Flag-FoxO1 at an MOI of 20. Cells were incubated overnight in glucose free and serum free medium. FoxO1 was immune-captured using anti-FLAG resin (Sigma). Protein samples were reduced with 10 mM Tris(2-carboxyethyl)phosphine hydrochloride, and alkylated with

20 mM chloroacetamide, then digested with trypsin in 2 M Urea at 25 °C overnight. The resulting peptides were separated on a nanoLC system and simultaneously detected in data-dependent analysis mode on an LTQ Orbitrap Fusion (Thermo Fisher Scientific).

By searching the resulting LCMSMS raw data against SwissProt Mouse protein database, the peptide and protein IDs were assigned using Mascot V2.5 (Matrix Science Inc.) on Proteome Discoverer 2.1 platform (Thermo Fisher Scientific). The WT and GCN5L1 LKO FoxO1 derived phosphopeptides were filtered out at 1% false discovery rate (FDR) and their relative abundances compared based on the areas under curve (AUC) of their corresponding chromatographic peaks. The representative spectra of MS analysis described this article, are shown as Supplementary Data 2.

**Immunofluorescence.** Cells were grown on coverslips, fixed with 4% paraformaldehyde and permeabilized with 0.1% Triton X-100 in PBS for 10 min. After blocking in 3% BSA for 1 h, the cells were incubated with the primary and corresponding fluorophore-conjugated secondary antibodies. Confocal images were captured with an LSM 710 confocal microscope. The mitochondria were determined by immunostaining with antibodies against Tom20 (Cell Signaling Technology). Nuclei were stained with DAPI (Molecular Probes).

**Quantitative real-time PCR.** Total RNA was prepared from cultured cells or mouse tissues using RNeasy purification kit (Qiagen). One microgram RNA of each sample was reverse transcribed with a cDNA reverse transcription kit (Invitrogen) and subjected to Q-PCR by using SYBR green mixture (Roche) with Roche Light-Cycler Q-PCR System. Primers were purchased from Qiagen.

**Measurement of reactive oxygen species in hepatocytes.** Total intracellular ROS levels and mitochondrial $H_2O_2$ levels were measured in primary hepatocytes cultured in glucose-free medium overnight. For intracellular ROS, cells were washed with PBS and incubated with 10 μM CM-H2DCFDA (Invitrogen) for 30 min at 37 C. Cells were trypsinized, and resuspended in PBS containing 1% BSA and 0.5 mM EDTA for FACS analysis using the LSRII instrument (BD). Mean fluorescence was measured and normalized to Con group with Veh treated or adenovirus vector infected. Mitochondrial $H_2O_2$ levels were measured using Amplex Red Hydrogen Peroxide Assay Kit (Invitrogen). Briefly, mitochondria from 3 million hepatocytes were sonicated in 1X reaction buffer with 1% NP-40, and centrifuged 10 min at 12,000 r.p.m. Mitochondrial lysates were collected for protein concentration and $H_2O_2$ measurement.

**Statistical analysis.** Sample sizes were sufficiently powered ($1-\beta = 0.8$, $\alpha = 0.05$) to detect at least a 20% difference in blood glucose levels. Results are displayed as the mean ± s.e.m. A comparison of groups was performed using two-tailed unpaired Student's $t$-test or one-way analysis of variance two-way ANOVA was used for quantification FoxO1 stability in the presence of CHX. A value of $P < 0.05$ was considered to indicate statistical significance.

**Data availability.** All data generated or analyzed during this study are included in this polished article and its Supplementary information files or available from the corresponding author on reasonable request.

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

## Acknowledgements

We thank Dr Oksana Gavrilova (NIDDK, Bethesda, MD, USA) for measurement of insulin and glucagon, Dr Ankit Saxena (FACS Core Facility, NHLBI, Bethesda, MD, USA) for assistance of FACS analysis, and Dr Marc Montmiry (The Salk Institute, La Jolla, CA, USA) for providing us adenovirus expression of CRE-luciferase reporter. This study was funded by NHLBI Division of Intramural Research funds to M.N.S.

## Author contributions

L.W. and M.N.S. designed and conceived the experiments. L.W., I.S., L.Z., K.W. and Y.C. performed and analyzed the experiments. K.H. and M.G. provided important experimental suggestions. L.Z., K.H. and M.N.S. wrote the manuscript.

## Additional information

**Competing interests:** The authors declare no competing financial interests.

