## [Peer Review File · Nature Communications]

Reviewers' Comments:

Reviewer #1 (Remarks to the Author)

Wang et al report that changes in mitochondrial acetylation has effects on hepatic glucose production, fasting glucose levels, levels of FoxO1 protein (due to effects on ubiquitylation and protein turnover), the expression of FoxO1 target genes, including G6Pase and PEPCK, and levels of other nuclear transcription factors, including HNF-4. This findings are based on studies in liver-specific GCN5L1 knockout mice, where the acetylation of multiple, yet unidentified, mitochondrial proteins appears to be is reduced based on western blotting of acetylated proteins using anti-acetyl lysine antibody, and fasting blood glucose levels are low. Studies in hepatocytes indicate that the abundance of G6Pase and PEPCK mRNA transcripts, gluconeogenesis, and FoxO1 protein stability and levels are decreased when GCN5L1 is knocked out and mitochondrial acetylation is low, and restoring mitochondrial (but not peroxisomal) GCN5L1 or knocking down Sirt3 and restoring mitochondrial acetylation reverses thee effects. Together, these studies reveal a novel relationship between mitochondrial acetylation status, FoxO1 protein turnover, gluconeogenic gene expression and glucose production in the liver.

The studies may be further strengthened by addressing the following points:

Major concerns:

1. A novel correlation between mitochondrial acetylation and FoxO1 levels (and corresponding change in G6Pase and PEPCK expression glucose production) is identified, but remains descriptive. Additional studies (or at least discussion) to explore the mechanisms linking mitochondrial acetylation and FoxO1 turnover would be helpful. Specifically, it is known that phosphorylation promotes ubiquitylation and degradation of FoxO1, and that acetylation of FoxO1 promotes its phosphorylation. Wang et al report that changes in mitochondrial acetylation has effects on hepatic glucose production, fasting glucose levels, levels of FoxO1 protein (due to effects on ubiquitylation and protein turnover), the expression of FoxO1 target genes, including G6Pase and PEPCK, and levels of other nuclear transcription factors, including HNF-4. This findings are based on studies in liver-specific GCN5L1 knockout mice, where the acetylation of multiple, yet unidentified, mitochondrial proteins appears to be is reduced based on western blotting of acetylated proteins using anti-acetyl lysine antibody, and fasting blood glucose levels are low. Studies in hepatocytes indicate that the abundance of G6Pase and PEPCK mRNA transcripts, gluconeogenesis, and FoxO1 protein stability and levels are decreased when GCN5L1 is knocked out and mitochondrial acetylation is low, and restoring mitochondrial (but not peroxisomal) GCN5L1 or knocking down Sirt3 and restoring mitochondrial acetylation reverses thee effects. Together, these studies reveal a novel relationship between mitochondrial acetylation status, FoxO1 protein turnover, gluconeogenic gene expression and glucose production in the liver.

The studies may be further strengthened by addressing the following points:

Major concerns:

1. A novel correlation between mitochondrial acetylation and FoxO1 levels (and corresponding change in G6Pase and PEPCK expression glucose production) is identified, but remains descriptive. Additional studies (or at least discussion) to explore the mechanisms linking mitochondrial acetylation and FoxO1 turnover would be helpful. Specifically, it is known that phosphorylation promotes ubiquitylation and degradation of FoxO1, and that acetylation of FoxO1 promotes its phosphorylation. Do changes in mitochondrial acetylation/function result in alteration in acetylation/phosphorylation/degradation of FoxO1? Studies to examine effects of the activity of pathways that are known to alter the phosphorylation and/or acetylation of FoxO1 may provide insight into mechanisms linking mitochondrial acetylation status to FoxO1 levels and glucose metabolism.

2. The physiological role of GCN5L1 in glucoregulation requires clarification, at least in discussion. Based on the authors previous studies (ref 3), the expression GCN5L1 is decreased in the liver during fasting, when acetylation of mitochondrial proteins and the function of FoxO proteins and hepatic glucose production are high, not low. This suggests that changes in GCN5L1 expression are not playing a major role in regulating FoxO1 activity and glucose metabolism in the liver in the response to fasting and feeding, and the manuscript (including the abstract which suggests that GCN5L1 is a nutrient-sensing regulatory protein) needs to be revised.

3. At the same time, the major point of the paper is that changes in mitochondrial acetylation are associated with changes in FoxO levels, glucose production and gene expression is novel and interesting. The data suggests that increased mitochondrial acetylation in fasting may be important in promoting hepatic glucose production, not just due to effects on mitochondrial energy metabolism, but also due to associated effects on FoxO protein stability and gene expression, and this can be stated more clearly in discussion. (Related: Previous studies indicate that production of intramitochondrial acetyl CoA can promote gluconeogenesis through allosteric effects on the activity of PCO and PDH. The present study suggests that intramitochondrial acetyl CoA can promote glucose production indirectly, due to changes in mitochondrial protein acetylation and subsequent effects on FoxO1 protein stability and G6Pase and PEPCCK gene expression.)

4. FoxO1 levels are low in hepatocytes from GCN5L1 KO mice, and may contribute to changes in G6Pase and PEPCCK mRNA levels. However, other results suggest that others factors also contribute to changes in gene expression and glucose metabolism in GCN5L1 KO mice. For example, FoxO1 suppresses the expression of glucokinase, yet glucokinase levels are not altered in GCN5L1 KO mice. Also, knocking out FoxO1 in the liver has little effect on 6 hr and 18 hr fasting glucose levels, yet glucose levels are low in GCN5L1 KO mice. Thus, it seems likely that other effects of knocking out GCN5L1 (and reducing acetylation of mitochondrial proteins) also contribute to changes in glucose metabolism and gene expression in these mice, not just changes in FoxO1 levels. Presumably, the mechanism(s) by which changes in mitochondrial protein acetylation alters FoxO1 proteins stability also exert other effects that contribute to changes in gene expression and glucose metabolism. This is supported by the finding that HNF-4 levels also are altered.

Other concerns to be addressed include:

1. Data on FoxO1 protein levels and G6Pase and PEPCCK gene expression are limited to hepatocytes in culture. These also need to be measured in the liver of WT and LKO mice.
2. Levels of both FoxO1 and HNF-4 are altered by changes in mitochondrial protein acetylation, and both contribute to G6Pase and PEPCCK gene expression. Therefore, the effects on glucose production are not due to effects on FoxO1 alone.
3. Related - what about PGC-1 α protein levels? (mRNA levels are shown, but not protein).
4. Also related - overexpression of FoxO1 rescues G6Pase and PEPCCK expression and glucose production in hepatocytes from GCN5L1 knockout hepatocytes. While we assume that these are direct effects of FoxO1, rescuing FoxO1 also may restores effects on mitochondrial protein acetylation, since FoxO1 can promote fatty acid oxidation in the liver.
5. P. 4 2nd line from bottom. Fasting glucose production was not measured in KO mice, just fasting glucose levels.
6. NAM inhibits multiple sirtuins, not just SIRT3, so that the authors cannot conclude that changes in the activity of other sirtuins may contribute to the effects of NAM treatment in their studies.
7. Insulin tolerance is not altered in GCN5L1 mice. Glucose tolerance and/or pyruvate tolerance tests would add further clarity to effects on glucose homeostasis.

changes in mitochondrial acetylation/function result in alteration in acetylation/phosphorylation/degradation of FoxO1?

2. The physiological role of GCN5L1 in glucoregulation requires clarification, at least in discussion. Based on the authors previous studies (ref 3), the expression GCN5L1 is decreased in the liver

during fasting, when acetylation of mitochondrial proteins and the function of FoxO proteins and hepatic glucose production are high, not low. This suggests that changes in GCN5L1 expression are not playing a major role in regulating FoxO1 activity and glucose metabolism in the liver in the response to fasting and feeding, and the manuscript (including the abstract which suggests that GCN5L1 is a nutrient-sensing regulatory protein) needs to be revised.

3. At the same time, the major point of the paper is that changes in mitochondrial acetylation are associated with changes in FoxO levels, glucose production and gene expression is novel and interesting. The data suggests that increased mitochondrial acetylation in fasting may be important in promoting hepatic glucose production, not just due to effects on mitochondrial energy metabolism, but also due to associated effects on FoxO protein stability and gene expression, and this can be stated more clearly in discussion. (Related: Previous studies indicate that production of intramitochondrial acetyl CoA can promote gluconeogenesis through allosteric effects on the activity of PCO and PDH. The present study suggests that intramitochondrial acetyl CoA can promote glucose production indirectly, due to changes in mitochondrial protein acetylation and subsequent effects on FoxO1 protein stability and G6Pase and PEPCK gene expression.)

4. FoxO1 levels are low in hepatocytes from GCN5L1 KO mice, and may contribute to changes in G6Pase and PEPCK mRNA levels. However, other results suggest that other factors also contribute to changes in gene expression and glucose metabolism in GCN5L1 KO mice. For example, FoxO1 suppresses the expression of glucokinase, yet glucokinase levels are not altered in GCN5L1 KO mice. Also, knocking out FoxO1 in the liver has little effect on 6 hr and 18 hr fasting glucose levels, yet glucose levels are low in GCN5L1 KO mice. Thus, it seems likely that other effects of knocking out GCN5L1 (and reducing acetylation of mitochondrial proteins) also contribute to changes in glucose metabolism and gene expression in these mice, not just changes in FoxO1 levels. Presumably, the mechanism(s) by which changes in mitochondrial protein acetylation alters FoxO1 protein stability also exert other effects that contribute to changes in gene expression and glucose metabolism. This is supported by the finding that HNF-4 levels also are altered.

Other concerns to be addressed include:

1. Data on FoxO1 protein levels and G6Pase and PEPCK gene expression are limited to hepatocytes in culture. These also need to be measured in the liver of WT and LKO mice.
2. Levels of both FoxO1 and HNF-4 are altered by changes in mitochondrial protein acetylation, and both contribute to G6Pase and PEPCK gene expression. Therefore, the effects on glucose production are not due to effects on FoxO1 alone.
3. Related - what about PGC-1 α protein levels? (mRNA levels are shown, but not protein).
4. Also related - overexpression of FoxO1 rescues G6Pase and PEPCK expression and glucose production in hepatocytes from GCN5L1 knockout hepatocytes. While we assume that these are direct effects of FoxO1, rescuing FoxO1 also may restore effects on mitochondrial protein acetylation, since FoxO1 can promote fatty acid oxidation in the liver.
5. P. 4 2nd line from bottom. Fasting glucose production was not measured in KO mice, just fasting glucose levels.
6. NAM inhibits multiple sirtuins, not just SIRT3, so that the authors cannot conclude that changes in the activity of other sirtuins may contribute to the effects of NAM treatment in their studies.
7. Insulin tolerance is not altered in GCN5L1 mice. Glucose tolerance and/or pyruvate tolerance tests would add further clarity to effects on glucose homeostasis.

Reviewer #2 (Remarks to the Author)

Overall, this is a thorough study that demonstrates GCN5L1 influences the acetylation status of FOXO1, thereby controlling its stability. The study could be moderately enhanced by the following points:

1. More molecular resolution. If GCN5L1 controls mitochondrial protein acetylation, then how does

it influence FOXO1 stability? Do the authors propose a role for GCN5L1 in the cytoplasm? If so, then how does mitochondrial GCN5L1 rescue the glycemic phenotype? This point was not clear. Also, the paper describes a key role for GCN5L1 to "facilitate" mitochondrial protein acetylation. Surprisingly, there are no protein acetylation studies in this manuscript. More work here would be supportive of the main point, and might help clarify the overall model.

2. The role for SIRT3 in this study is unclear. While the simple yin-yang of SIRT3-GCN5L1 model might be a useful mental construct, the role of SIRT3 in this study is entirely unexplored and quite frankly underwhelming. The authors might consider omitting these data, which would allow more emphasis on the key point of the paper: that GCN5L1 influences hepatic gluconeogenesis.

Reviewer #3 (Remarks to the Author)

In this manuscript, the authors provide the role of GCN5L1 in the control of hepatic gluconeogenesis. They suggested that the depletion of mitochondrial acetyltransferase GCN5L1 reduced expression of hepatic gluconeogenic genes by affecting protein stability of FoxO1. While the potential connection between the mitochondrial acetylation and hepatic gluconeogenesis is interesting, several key data are missing to support the authors' suggestions in the manuscript. Please refer to the following specific comments.

Major comments:

1. To claim that the depletion of GCN5L1 reduces hepatic gluconeogenesis, the authors should measure gluconeogenic flux. At the very least, the authors should perform pyruvate challenge test by using WT and GCN5L1 LKO mice.
2. The authors should also report whether the depletion of GCN5L1 affects mitochondrial integrity in the liver, since defects in the mitochondria can easily affect glucose metabolism
3. More importantly, the authors should provide the exact mechanism by which the depletion of mitochondrial acetyltransferase affects FoxO1 stability. Authors should also map the key lysine residues that are targets for the ubiquitylation.
4. The authors should then express either WT or mutant (lysine residues) FoxO1 in GCN5L1 KO hepatocytes (by using adenovirus system) to verify whether expression of mutant FoxO1 can rescue the glucose phenotype shown in GCN5L1 knockout.
5. What is the genetic background for WT and KO mice. Plasma insulin levels are too high for 6h-fasted mice.
6. To fully determine the hepatic insulin responsiveness, the authors should measure insulin signaling by western blot analysis (p-Akt, p-IR) following injection of a bolus of insulin in mice.

Minor comments:

1. Primary hepatocytes are extremely hard to transfect. What is the transfection efficiency of FoxO1 transfection in Fig 6f?
2. Along the same line, the authors should report the protein levels of SIRT3 upon siRNA-mediated knockdown (Fig 7g).

Reviewers' comments:

Reviewer #1 (expert in gluconeogenesis/foxo1) (Remarks to the Author):

Wang et al report that changes in mitochondrial acetylation has effects on hepatic glucose production, fasting glucose levels, levels of FoxO1 protein (due to effects on ubiquitylation and protein turnover), the expression of FoxO1 target genes, including G6Pase and PEPCK, and levels of other nuclear transcription factors, including HNF-4. This findings are based on studies in liver-specific GCN5L1 knockout mice, where the acetylation of multiple, yet unidentified, mitochondrial proteins appears to be reduced based on western blotting of acetylated proteins using anti-acetyl lysine antibody, and fasting blood glucose levels are low. Studies in hepatocytes indicate that the abundance of G6Pase and PEPCK mRNA transcripts, gluconeogenesis, and FoxO1 protein stability and levels are decreased when GCN5L1 is knocked out and mitochondrial acetylation is low, and restoring mitochondrial (but not peroxisomal) GCN5L1 or knocking down Sirt3 and restoring mitochondrial acetylation reverses these effects. Together, these studies reveal a novel relationship between mitochondrial acetylation status, FoxO1 protein turnover, gluconeogenic gene expression and glucose production in the liver.

The studies may be further strengthened by addressing the following points:

Major concerns:

1. A novel correlation between mitochondrial acetylation and FoxO1 levels (and corresponding change in G6Pase and PEPCK expression glucose production) is identified, but remains descriptive. Additional studies (or at least discussion) to explore the mechanisms linking mitochondrial acetylation and FoxO1 turnover would be helpful. Specifically, it is known that phosphorylation promotes ubiquitylation and degradation of FoxO1, and that acetylation of FoxO1 promotes its phosphorylation. Do changes in mitochondrial acetylation/function result in alteration in acetylation/phosphorylation/degradation of FoxO1? Studies to examine effects of the activity of pathways that are known to alter the phosphorylation and/or acetylation of FoxO1 may provide insight into mechanisms linking mitochondrial acetylation status to FoxO1 levels and glucose metabolism.

Thank you for this excellent suggestion. We investigated this further by initially evaluating whether the phosphorylation status of Foxo1 was altered in the GCN5L1 null hepatocytes compared to wildtype controls using LCMSMS. In the absence of GCN5L1 Foxo1 showed excess phosphorylation of multiple serine and threonine residues. The sites identified were found to be canonical p38 and ERK targets, although only the inhibition of ERK restored Foxo1 protein levels, gluconeogenic gene expression and glucose production. Moreover, the phosphorylation of both p42 and p44 ERK isoforms showed higher phosphorylation in GCN5L1 LKO cells and the reconstitution of both the mitochondrial targeted and wildtype GCN5L1 reversed excessive ERK phosphorylation in the LKO cells. Putting these data together shows that the absence of GCN5L1 in the mitochondria does initiate retrograde signaling to activate ERK, which in turn phosphorylates Foxo1. In keeping with the cross-talk between protein phosphorylation and ubiquitylation, this points to a novel initiation of retrograde signaling to promote Foxo1 turnover, with the resultant effect on blunting gluconeogenesis.

2. The physiological role of GCN5L1 in glucoregulation requires clarification, at least in discussion. Based on the authors previous studies (ref 3), the expression GCN5L1 is decreased in the liver during fasting, when acetylation of mitochondrial proteins and the function of FoxO proteins and hepatic glucose production are high, not low. This suggests that changes in GCN5L1 expression are not playing a major

role in regulating FoxO1 activity and glucose metabolism in the liver in the response to fasting and feeding, and the manuscript (including the abstract which suggests that GCN5L1 is a nutrient-sensing regulatory protein) needs to be revised.

Thank you for this thoughtful comment and we agree that this concept is not delineated comprehensively enough to support the title and some of the discussion in the manuscript. To correct this, the title has been changed from 'Mitochondrial acetylation sensing modulates Foxo1 levels to control hepatocyte gluconeogenesis' to "GCN5L1 modulates cross talk between mitochondria and cell signaling in the control of gluconeogenesis". Additionally, all references to nutrient sensing has been removed from the manuscript and the abstract, introduction and discussion have all been modified to reflect this and are shown in the track changes version of the manuscript.

3. At the same time, the major point of the paper is that changes in mitochondrial acetylation are associated with changes in FoxO levels, glucose production and gene expression is novel and interesting. The data suggests that increased mitochondrial acetylation in fasting may be important in promoting hepatic glucose production, not just due to effects on mitochondrial energy metabolism, but also due to associated effects on FoxO protein stability and gene expression, and this can be stated more clearly in discussion. (Related: Previous studies indicate that production of intra-mitochondrial acetyl CoA can promote gluconeogenesis through allosteric effects on the activity of PCO and PDH. The present study suggests that intramitochondrial acetyl CoA can promote glucose production indirectly, due to changes in mitochondrial protein acetylation and subsequent effects on FoxO1 protein stability and G6Pase and PEPCK gene expression.)

Thank you for this comment. We have added text in the discussion to point out that this effect of GCN5L1 does appear to be an indirect effect from the acetyl-modifications in the mitochondria. This was expanded on in the discussion in the middle paragraph on page 13 and the subsequent paragraph that extends onto page 14 of the manuscript.

4. FoxO1 levels are low in hepatocytes from GCN5L1 KO mice, and may contribute to changes in G6Pase and PEPCK mRNA levels. However, other results suggest that other factors also contribute to changes in gene expression and glucose metabolism in GCN5L1 KO mice. For example, FoxO1 suppresses the expression of glucokinase, yet glucokinase levels are not altered in GCN5L1 KO mice. Also, knocking out FoxO1 in the liver has little effect on 6 hr and 18 hr fasting glucose levels, yet glucose levels are low in GCN5L1 KO mice. Thus, it seems likely that other effects of knocking out GCN5L1 (and reducing acetylation of mitochondrial proteins) also contribute to changes in glucose metabolism and gene expression in these mice, not just changes in FoxO1 levels. Presumably, the mechanism(s) by which changes in mitochondrial protein acetylation alters FoxO1 proteins stability also exert other effects that contribute to changes in gene expression and glucose metabolism. This is supported by the finding that HNF-4 levels also are altered.

We agree that the pathway identified in this study is probably not the only pathway involved and we have added a section to the discussion pointing out the difference in our data to that found with the previously published liver specific Foxo1 knockout mouse. This is added to the middle paragraph on page 14. At the same time, we discuss potential additional mitochondrial and acetylation events that may or may not be operational as contributory components in the composite glucose production. The HNF-4 immunoblot was removed, as we have not explored this in any detail and think it would be more prudent not to introduce a concept that has not been evaluated further in this current manuscript.

Other concerns to be addressed include:

1. Data on FoxO1 protein levels and G6Pase and PEPCK gene expression are limited to hepatocytes in culture. These also need to be measured in the liver of WT and LKO mice. Thank you for the suggestion. We have added these data in Supplementary Fig 3a, 3b.
2. Levels of both FoxO1 and HNF-4 are altered by changes in mitochondrial protein acetylation, and both contribute to G6Pase and PEPCK gene expression. Therefore, the effects on glucose production are not due to effects on FoxO1 alone. – *Yes we acknowledge this and have added a section in the discussion where we sight the published response to liver-specific Foxo1 depletion, which also suggests that this is but one component in the overall regulation of gluconeogenesis.* This is discussed in the middle of page 14, and is highlighted in the track changes version of the revision.
3. Related - what about PGC-1 α protein levels? (mRNA levels are shown, but not protein). – *The endogenous protein levels of PGC-1 α are very low and we were not able to detect using our antibodies by Western Blot analysis.*
4. Also related - overexpression of FoxO1 rescues G6Pase and PEPCK expression and glucose production in hepatocytes from GCN5L1 knockout hepatocytes. While we assume that these are direct effects of FoxO1, rescuing FoxO1 also may restores effects on mitochondrial protein acetylation, since FoxO1 can promote fatty acid oxidation in the liver. *Thank you for this suggestion. We did explore the effect of Foxo1 overexpression of mitochondrial protein acetylation and we did not see any discernable effect. The antibodies that recognize acetylated lysine residue's may not have the sensitivity to elucidate differences on specific proteins and so this possibility cannot be excluded.*
5. P. 4 2nd line from bottom. Fasting glucose production was not measured in KO mice, just fasting glucose levels. – *This has been corrected and the data from the pyruvate challenge test is shown in Fig. 2f and in Supplemental Fig. 2c.*
6. NAM inhibits multiple sirtuins, not just SIRT3, so that the authors cannot conclude that changes in the activity of other sirtuins may contribute to the effects of NAM treatment in their studies. – *Multiple reviewers had concerns about the addition of the sirtuin studies to this manuscript and we agree that the specificity of the inhibitor precludes our attribution of its effects to SIRT3. Hence, these data have all been removed.*
7. Insulin tolerance is not altered in GCN5L1 mice. Glucose tolerance and/or pyruvate tolerance tests would add further clarity to effects on glucose homeostasis. *Thank you for this suggestion, these tests have now been performed and show no difference in glucose tolerance but a modest, but significant blunting of glucose production in response to pyruvate. This was performed in two separate studies using 16 LKO and 18 wildtype mice and the data showed a consistent blunting of glucose production in the GCN5L1 LKO mice.*

Reviewer #2 (expert in sirtuins) (Remarks to the Author):

Overall, this is a thorough study that demonstrates GCN5L1 influences the acetylation status of FOXO1, thereby controlling it's stability. The study could be moderately enhanced by the following points:

1. More molecular resolution. If GCN5L1 controls mitochondrial protein acetylation, then how does it influence FOXO1 stability? Do the authors propose a role for GCN5L1 in the cytoplasm? If so, then how does mitochondrial GCN5L1 rescue the glycemic phenotype? This point was not clear. Also, the paper describes a key role for GCN5L1 to "facilitate" mitochondrial protein acetylation. Surprisingly, there are no protein acetylation studies in this manuscript. More work here would be supportive of the main point, and might help clarify the overall model.

Thank you for this very salient critique. We did not find any significant or difference in the extent of Foxo1 acetylation following the overexpression of Flag-tagged FOxo1, Flag immunoprecipitation and immunoblot analysis with an antibody directed against acetylated lysine residues. Hence, to expand the molecular underpinning of the substantial effect of GCN5L1 levels on Foxo1 stability and its effects on gluconeogenesis we evaluated Foxo1 phosphorylation. This avenue was pursued given that protein phosphorylation has been found to facilitate protein ubiquitylation and degradation on other proteins. We investigated this further by initially evaluating whether the phosphorylation status of Foxo1 was altered in the GCN5L1 null hepatocytes compared to wildtype controls using LCMSMS. In the absence of GCN5L1 Foxo1 showed excess phosphorylation of multiple serine and threonine residues. The sites identified were found to be canonical p38 and ERK targets, although only the inhibition of ERK restored Foxo1 protein levels, gluconeogenic gene expression and glucose production. Moreover, the phosphorylation of both p42 and p44 ERK isoforms showed higher phosphorylation in GCN5L1 LKO cells and the reconstitution of both the mitochondrial targeted and wildtype GCN5L1 reversed excessive ERK phosphorylation in the LKO cells. Putting these data together shows that the absence of GCN5L1 in the mitochondria does initiate retrograde signaling to activate ERK, which in turn phosphorylates Foxo1. In keeping with the cross-talk between protein phosphorylation and ubiquitylation, this points to a novel initiation of retrograde signaling to promote Foxo1 turnover, with the resultant effect on blunting gluconeogenesis.

To your second concern pertaining to acetylation. We agree that this manuscript has not identified which proteins in the mitochondria have a differential acetylation effect to orchestrate the findings observed. To direct the emphasis of the manuscript to the work described, the title has been changed from 'Mitochondrial acetylation sensing modulates Foxo1 levels to control hepatocyte gluconeogenesis' to "GCN5L1 modulates cross talk between mitochondria and cell signaling in the control of gluconeogenesis".

2. The role for SIRT3 in this study is unclear. While the simple yin-yang of SIRT3-GCN5L1 model might be a useful mental construct, the role of SIRT3 in this study is entirely unexplored and quite frankly underwhelming. The authors might consider omitting these data, which would allow more emphasis on the key point of the paper: that GCN5L1 influences hepatic gluconeogenesis.

Thank you for this suggestion. The SIRT3 and NAM data have been removed from this study and replaced with the new data generated showing that the depletion of GCN5L1 augments ERK activation and Foxo1 phosphorylation as a component of the GCN5L1 level dependent effects on gluconeogenesis.

Reviewer #3 (expert in gluconeogenesis/foxo1)(Remarks to the Author):

In this manuscript, the authors provide the role of GCN5L1 in the control of hepatic gluconeogenesis. They suggested that the depletion of mitochondrial acetyltransferase GCN5L1 reduced expression of hepatic gluconeogenic genes by affecting protein stability of FoxO1. While the potential connection between the mitochondrial acetylation and hepatic gluconeogenesis is interesting, several key data are missing to support the authors' suggestions in the manuscript. Please refer to the following specific comments.

Major comments:

1. To claim that the depletion of GCN5L1 reduces hepatic gluconeogenesis, the authors should measure

gluconeogenic flux. At the very least, the authors should perform pyruvate challenge test by using WT and GCN5L1 LKO mice.

Thank you this has been addressed and the data showing the pyruvate challenge test is now shown in Fig. 2f. and supplementary Fig. 2c.

2. The authors should also report whether the depletion of GCN5L1 affects mitochondrial integrity in the liver, since defects in the mitochondria can easily affect glucose metabolism.

This has also been addressed and we do not find any major changes in mitochondrial function at the levels of oxygen consumption, mitochondrial genomic DNA levels or at the level of mitochondrial protein levels. These data are not shown in the manuscript but are reviewed in the discussion section in the middle paragraph on page 13 and the data is shown here.

3. More importantly, the authors should provide the exact mechanism by which the depletion of mitochondrial acetyltransferase affects FoxO1 stability. Authors should also map the key lysine residues that are targets for the ubiquitylation.

The mechanism whereby a change in an intra-mitochondrial post-translational modification, deacetylation in this case, modifies a cytosolic/nuclear located transcription factor is very challenging. At the same time, we do not find a difference in acetylation of Foxo1 in the WT and LKO hepatocytes. However, the extent of mitochondrial protein acetylation has been found to regulate transcriptional mechanisms via retrograde signaling from the mitochondria. This concept is now expanded upon in the discussion of the manuscript (final paragraph on page 14 onto page 15 of the manuscript). At the same time we have explored this retrograde signaling concept in greater detail and explored the effect of GCN5L1 depletion on Foxo1 phosphorylation. This avenue was pursued given that protein phosphorylation has been found to facilitate protein ubiquitylation and degradation on other proteins. We investigated this further by initially evaluating whether the phosphorylation status of Foxo1 was altered in the GCN5L1 null hepatocytes compared to wildtype controls using LCMSMS. In the absence of GCN5L1 Foxo1 showed excess phosphorylation of multiple serine and threonine residues. The sites identified were found to be canonical p38 and ERK targets, although only the inhibition of ERK restored Foxo1 protein levels, gluconeogenic gene expression and glucose production. Moreover, the

phosphorylation of both p42 and p44 ERK isoforms showed higher phosphorylation in GCN5L1 LKO cells and the reconstitution of both the mitochondrial targeted and wildtype GCN5L1 reversed excessive ERK phosphorylation in the LKO cells. Putting these data together shows that the absence of GCN5L1 in the mitochondria does initiate retrograde signaling to activate ERK, which in turn phosphorylates Foxo1. In keeping with the cross-talk between protein phosphorylation and ubiquitylation, this points to a novel initiation of retrograde signaling to promote Foxo1 turnover, with the resultant effect on the blunting of gluconeogenesis.

At the same time the concept that perturbations in mitochondrial homeostasis can effect cytosolic signaling is increasingly been recognized. This is cited in the study with respect to the extra-mitochondrial effects on regulatory proteins by changes in mitochondrial acetylation. In addition, a new study published in the December 7th issue of Science Translational Medicine shows that the inhibition of the mitochondrial pyruvate carrier inhibits mTOR signaling with effects on autophagy. Here too, the intra-mitochondrial mechanism has yet to be determined.

4. The authors should then express either WT or mutant (lysine residues) FoxO1 in GCN5L1 KO hepatocytes (by using adenovirus system) to verify whether expression of mutant FoxO1 can rescue the glucose phenotype shown in GCN5L1 knockout.

Given that we found that the phosphorylation of multiple serine and threonine residues of Foxo1 were modified by ERK signaling and given that ERK activation appears upstream of the Foxo1 effect, we feel that the studies going forward should focus on the ERK activation to enable us to better understand the role of GCN5L1 in retrograde signaling. Hence the mutations of the Foxo1 protein were not undertaken.

5. What is the genetic background for WT and KO mice. Plasma insulin levels are too high for 6h-fasted mice.

The mice are C57BL6 and the Insulin levels were repeated in the fed and after an overnight fast. The revised data suggest that the 6h fast data from our prior experiments were more closely aligned to the fed levels. Whether this reflects the rate of food digestion may be a possibility. Nevertheless, the 6h data has been removed and replaced with the newer ON fast and fed data in Fig. 2c.

6. To fully determine the hepatic insulin responsiveness, the authors should measure insulin signaling by western blot analysis (p-Akt, p-IR) following injection of a bolus of insulin in mice.

This was performed and the findings were quite surprising and suggest that this GCN5L1 depletion effect on gluconeogenesis is independent on insulin signaling. In brief, there was no difference in the phosphorylation of the IR and there appeared to be a modest reduction in the relative phosphorylation of Akt in parallel with a modest increase in total Akt levels in the LKO. As reduced gluconeogenesis would normally be associated with improved insulin sensitivity – hence our suggestion that this GCN5L1 depletion effect is independent of insulin signaling.

Minor comments:

1. Primary hepatocytes are extremely hard to transfect. What is the transfection efficiency of FoxO1 transfection in Fig 6f? – *The primary hepatocytes were transfected with a Flag-tagged Foxo1 and immunofluorescence using a primary Flag antibody showed an approximate 40% transfection efficiency. This study was repeated using an adenoviral vector harboring Foxo1 and the results obtained were the same as with the overexpression system.*

2. Along the same line, the authors should report the protein levels of SIRT3 upon siRNA-mediated knockdown (Fig 7g). – *Due to the questions of the specificity of this investigation of SIRT3 by numerous reviewers of this study, these experiments have now been removed.*

Reviewers' Comments:

Reviewer #1 (Remarks to the Author)

This revised manuscript by Wang et al provides novel evidence linking GCN5L1-dependent effects on mitochondrial protein acetylation to changes in ERK signaling, phosphorylation and turnover of FoxO1, changes in PEPCK and G6Pase expression, and glucose production in hepatocytes. These findings provided new insight into mechanisms linking mitochondrial function to the regulation of intracellular signaling pathways, gene expression and glucose metabolism in the liver. The manuscript can be further improved strengthened by addressing the following concerns:

1. Title. One important finding of this paper is the connection between changes in mitochondrial function and FoxO1 stability/turnover, and it would be helpful to indicate this in the title, e.g. by inserting "Foxo1 stability and..." just before gluconeogenesis.
2. The physiological relevance of the results of the paper are not entirely clear. If suppression of GCN5L1 promotes reduced acetylation of mitochondrial proteins and increased turnover of Foxo1, this would be adaptive to the fed state: is GCN5L1 suppressed in the liver of fed vs. fasted mice (or increased in fasted vs fed mice)? If not, then it is not clear that changes in GCN5L1 expression contribute to changes in glucose production in the transition between from fed to fasted states. Nevertheless, GCN5L1 may still play an important role in promoting acetylation of mitochondrial proteins that are important for promoting changes in Foxo1 protein stability, levels and activity under fasting conditions. This could be addressed in discussion.
3. p. 2. Abstract. First sentence. It is not clear from the way it is stated that this sentence refers to published results, and is not a finding of this report. If allowed, providing a reference would clarify this point. If not allowed, then the authors might indicate that this has been previously reported.
4. p. 2, line 51. "transcriptional profiling" suggests that gene array or RNA seq studies were done. Instead, it seems that targeted gene expression studies were performed to examine changes in glycolytic, glycogenic and gluconeogenic pathways. Please clarify.
5. p. 2, line 55. should be "reversed" – past tense.
6. p. 2, lines 58-59. Delete "The". Start with "Reconstitution..."
7. p. 2, line 61. Delete "levels".
8. p. 3, line 66. Delete hyphen after "substrate".
9. p. 3, lines 72-73. Why is this "in contrast". Seems "In addition" is more appropriate.
4. p. 4, lines 90-95. This is an interesting hypothesis, but it is not examined in this paper, and so probably best left unsaid. Changes in acetylation of mitochondrial proteins are examined – but not "acetyl CoA" sensing. (Related to this, it would be interesting how mitochondrial protein acetylation is altered. For example, is beta oxidation impaired? Is SIRT3 expression/activity altered? Are levels of acetyl-CoA altered? Is pyruvate carboxylase activity altered independently of changes in acetyl-CoA?).
5. p. 4, line 97. Should read "...dysregulation..."
6. p. 4, lines 97-98. Hyperglycemia is a sine qua non of diabetes, but not insulin resistance (e.g. in metabolic syndrome, glucose levels are normal but subjects are insulin resistance. Can also delete

the sentence beginning "Since excessive..."

7. p. 4, line 104. Delete "hepatic" – since yeast do not have a liver.

8. p. 4, lines 108-109. The repeated reference to acetyl CoA sensing seems overdone. The paper by Perry et al related gluconeogenesis to activation of pyruvate carboxylase by acetyl CoA produced by fatty acid oxidation, which also provides ATP and NADH required for glucose production. Here, the authors are not looking at changes in fatty acid oxidation and acetyl CoA levels (which would be interesting to do – see above), just acetylation, which may be regulated at other levels, including changes in acetyltransferase and deacetylase activity.

9. p. 5, line 110, and Fig 2f and supplementary Fig 2c. The results of pyruvate challenge testing need further evaluation (below). When evaluating the results of pyruvate challenge tests, it is important to take into account differences in glucose levels at time 0. Since fasting glucose levels are low in LKO mice (Fig 2c and Suppl Fig 2c), post-pyruvate glucose levels may also be lower due to differences in baseline – not because less glucose is made from the pyruvate that is given. Baseline glucose levels at time 0 need to be subtracted from glucose levels at subsequent time points to estimate pyruvate-dependent changes in glucose. Results need to be re-analyzed after subtracting out glucose levels at time 0 for each of the mice, and the results should be presented both as curves (as in Fig 2c, but for all mice combined) and area under the curve.

10. p. 6, line 151 and supplementary Fig 2d. The amount of phosphor-Akt does not appear to be low. The reduction in p-Akt/total Akt seems to reflect an increase in total Akt, but not a reduction in phosphor-Akt. This would suggest that although the phosphorylation of Akt may be relatively inefficient, the level of active Akt is not reduced, and signaling to downstream targets may be intact, including, e.g. GSK-3 and FoxO proteins. This might be taken into consideration when discussing factors that promote increased degradation to FoxO1 – since phosphorylation of FoxO1 is known to promote ubiquitylation and degradation by the proteasome.

11. p. 7, line 159. Pyruvate carboxylase is abbreviated PC elsewhere – it would be helpful to insert the abbreviation here for clarity, and to explain other abbreviations when they occur elsewhere.

12. p. 7, lines 163-165. It doesn't seem correct to state that PEPCK and G6Pase transcript levels were similarly affected in vivo in liver and in hepatocytes from LKO mice. For example, PEPCK transcripts were significantly (~70%) decreased in LKO hepatocytes, but were only reduced by ~20% in vivo, and this effect was not statistically significant. This more limited effect should be acknowledged, and not buried as a supplemental figure. It also could be mentioned in discussion.

13. p. 8, line 179. Should be "glucagon" instead of "glycogen".

14. p. 7-8, lines 178-179. The point that basal glucose production is reduced in LKO, but the ability of glucagon to stimulate (fold-increase) glucose production is intact can, and should be stated more clearly. This is an important point, and warrants an additional bar graph to show that glucagon stimulated fold-increase is not dependent on GCN5L1, in my opinion.

15. p. 8, lines 199-200. The data shown in Supplementary Figs 4b and 4c is important since it provides direct evidence that mitochondrial GCN5L1 is positively linked to gluconeogenic gene expression glucose production, and would be best shown within Fig 6.

16. p. 8, line 199 and Fig 6b. The organization of the figures in Fig 6 is confusing, with 6b located below 6a, and then 6c located on the same level of 6a. I suggest showing 6b, together with the data in supplemental figures 4a and 4b, before showing studies related to Foxo1 turnover.

17. p. 9, line 203-205. The data presented indicate that Mt-GCN5L1 plays an important role in promoting or maintaining PEPCK and G6Pase expression and glucose production in hepatocytes –

but this is not the same as "regulation", e.g. in response to fasting or feeding.

18. p. 9, line 216. It is important to know what level of FoxO1 protein was expressed. If FoxO1 expression is more than recovered and it is markedly overexpressed, then the recovery of gene expression and glucose production may reflect an over compensation due to very high levels of Foxo1. Put another way, this recovery experiment does not rule out the possibility that other factors also may contribute to reduced PEPCCK and G6Pase expression and glucose production in LKO cells, and this should be noted in the text and/or discussion.

19. p. 9, line 224. Should read "differentially" instead of "differentiated".

20. p. 20. Line 247. Suggest ERK "dependent" instead of "mediated".

21. p. 12, lines 288-299. Awkward. "although" occurs twice in this sentence.

22. p. 13. Line 294. Suggest - Insert "with" after "in contrast".

23. p. 13, line 310. Suggest "did" instead of "was".

Reviewer #2 (Remarks to the Author)

The authors have satisfactorily addressed my previous concerns. Their focus on GCN5L1 and one mechanism by which its ablation controls FOXO1 stability, along with their removal of the sirtuin data, together serve to focus the manuscript. Their initial characterization of liver-specific GCN5L1KO mice is interesting, and stimulates several future studies.

Reviewer #3 (Remarks to the Author)

In this revised manuscript, the authors added more mechanism by which hepatic depletion of GCN5L1 affects Foxo1 activity and gluconeogenesis. They now suggested that increased Erk activity was observed, and is the key regulator for the metabolic phenotype in GCN5L1 liver-specific KO mice. They also did not find any changes in mitochondrial function, so they don't have any clue how the depletion of GCN5L1 can affect cytosolic signaling and, as a result, hepatic gluconeogenesis. Without the clear-cut molecular mechanism linking any changes in mitochondrial events (that are direct targets of GCN5L1) and ERK-Foxo1 pathway, this study is more of a speculation and a description of their findings in non-physiological setting. As shown in their response to reviewer 1's question, the authors also admitted that GCN5L1 can no longer be considered as a nutrient sensor since its expression pattern does not match with Foxo1 activity. This significantly dampens the reviewer's enthusiasm to recommend the manuscript for publication.

Reviewer #1 (Remarks to the Author):

This revised manuscript by Wang et al provides novel evidence linking GCN5L1-dependent effects on mitochondrial protein acetylation to changes in ERK signaling, phosphorylation and turnover of FoxO1, changes in PEPCK and G6Pase expression, and glucose production in hepatocytes. These findings provided new insight into mechanisms linking mitochondrial function to the regulation of intracellular signaling pathways, gene expression and glucose metabolism in the liver. The manuscript can be further improved strengthened by addressing the following concerns:

Thank you for your review of our manuscript. We have responded to each of your comments below and a version of the revised manuscript with track changes is included for your review.

1. Title. One important finding of this paper is the connection between changes in mitochondrial function and FoxO1 stability/turnover, and it would be helpful to indicate this in the title, e.g. by inserting “Foxo1 stability and...” just before gluconeogenesis.

Thank you for this suggestion, the title has been changed to: GCN5L1 modulates cross-talk between mitochondria and cell signaling to regulate FoxO1 stability and gluconeogenesis

2. The physiological relevance of the results of the paper are not entirely clear. If suppression of GCN5L1 promotes reduced acetylation of mitochondrial proteins and increased turnover of Foxo1, this would be adaptive to the fed state: is GCN5L1 suppressed in the liver of fed vs. fasted mice (or increased in fasted vs fed mice)? If not, then it is not clear that changes in GCN5L1 expression contribute to changes in glucose production in the transition between from fed to fasted states. Nevertheless, GCN5L1 may still play an important role in promoting acetylation of mitochondrial proteins that are important for promoting changes in Foxo1 protein stability, levels and activity under fasting conditions. This could be addressed in discussion.

Thank you for pointing this out. We have not expanded on this concept extensively in this study, but we do show that GCN5L1 levels are induced after a six hour fast, which correlates with the more robust differential (blunting) of glucose levels in the GCN5L1 LKO mice. This is discussed in the results section and shown in Supplementary Figure 2a and shown here.

This concept is also included in the discussion section on page 13, line 15 onwards. Where we point out that this is an interesting observation, that needs to be explored further.

3. p. 2. Abstract. First sentence. It is not clear from the way it is stated that this sentence refers to published results, and is not a finding of this report. If allowed, providing a reference would clarify this point. If not allowed, then the authors might indicate that this has been previously reported.

Thank you we have corrected this and clearly state that this was previously found.

4. p. 2, line 51. “transcriptional profiling” suggests that gene array or RNA seq studies were done. Instead, it seems that targeted gene expression studies were performed to examine changes in glycolytic, glycogenic and gluconeogenic pathways. Please clarify.

This has been corrected to state that targeted gene expression studies were performed.

5. p. 2, line 55. should be “reversed” – past tense.

This sentence was removed from the abstract to enable the additional data on ROS signaling.

6. p. 2, lines 58-59. Delete “The”. Start with “Reconstitution...”

This has been changed.

7. p. 2, line 61. Delete “levels”.

This has been deleted.

8. p. 3, line 66. Delete hyphen after “substrate”.

This has been deleted.

9. p. 3, lines 72-73. Why is this “in contrast”. Seems “In addition” is more appropriate.

This has been corrected.

4. p. 4, lines 90-95. This is an interesting hypothesis, but it is not examined in this paper, and so probably best left unsaid. Changes in acetylation of mitochondrial proteins are examined – but not “acetyl CoA” sensing. (Related to this, it would be interesting how mitochondrial protein acetylation is altered. For example, is beta oxidation impaired? Is SIRT3 expression/activity altered? Are levels of acetyl-CoA altered? Is pyruvate carboxylase activity altered independently of changes in acetyl-CoA?).

The discussion of this concept has been removed. Now that we have found that mitochondrial ROS levels are altered by the depletion of GCN5L1 future studies will explore the mechanism/s underpinning this and these may in part result from alterations in enzyme activities within mitochondria. However, as the investigation of these concepts will be a very substantial new area of study, we feel this is less pertinent to the signaling pathways explored in this manuscript.

5. p. 4, line 97. Should read “...dysregulation...”

Corrected, thank you.

6. p. 4, lines 97-98. Hyperglycemia is a sine qua non of diabetes, but not insulin resistance (e.g. in metabolic syndrome, glucose levels are normal but subjects are insulin resistance. Can also delete the sentence beginning “Since excessive...”

Thank you for pointing this out, the reference to insulin resistance has been removed from both this introductory section and from the discussion.

7. p. 4, line 104. Delete “hepatic” – since yeast do not have a liver.

Corrected, thank you.

8. p. 4, lines 108-109. The repeated reference to acetyl CoA sensing seems overdone. The paper by Perry et al related gluconeogenesis to activation of pyruvate carboxylase by acetyl CoA produced by fatty acid oxidation, which also provides ATP and NADH required for glucose production. Here, the authors are not looking at changes in fatty acid oxidation and acetyl CoA levels (which would be interesting to do – see above), just acetylation, which may be regulated at other levels, including changes in acetyltransferase and deacetylase activity.

The discussion of this concept has been removed from the introduction.

9. p. 5, line 110, and Fig 2f and supplementary Fig 2c. The results of pyruvate challenge testing need further evaluation (below). When evaluating the results of pyruvate challenge tests, it is important to take into account differences in glucose levels at time 0. Since fasting glucose levels are low in LKO mice (Fig 2c and Suppl Fig 2c), post-pyruvate glucose levels may also be lower due to differences in baseline – not because less glucose is made from the pyruvate that is given. Baseline glucose levels at time 0 need to be subtracted from glucose levels at subsequent time points to estimate pyruvate-dependent changes in glucose. Results need to be re-analyzed after subtracting out glucose levels at time 0 for each of the mice, and the results should be presented both as curves (as in Fig 2c, but for all mice combined) and area under the curve.

Thank you for pointing this out we have corrected this and displayed the data as the new figures 2f, g and shown below.

10. p. 6, line 151 and supplementary Fig 2d. The amount of phosphor-Akt does not appear to be low. The reduction in p-Akt/total Akt seems to reflect an increase in total Akt, but not a reduction in phosphor-Akt. This would suggest that although the

phosphorylation of Akt may be relatively inefficient, the level of active Akt is not reduced, and signaling to downstream targets may be intact, including, e.g. GSK-3 and FoxO proteins. This might be taken into consideration when discussing factors that promote increased degradation to FoxO1 – since phosphorylation of FoxO1 is known to promote ubiquitylation and degradation by the proteasome.

We agree there is no substantial change in insulin signaling and this has been clarified in the manuscript. This has been stated in the results section on page 7, last line of paragraph 2 and is stated in the final sentence of the first paragraph of the discussion.

11. p. 7, line 159. Pyruvate carboxylase is abbreviated PC elsewhere – it would be helpful to insert the abbreviation here for clarity, and to explain other abbreviations when they occur elsewhere.

This has been corrected and we believe we have clarified the additional abbreviations.

12. p. 7, lines 163-165. It doesn't seem correct to state that PEPCK and G6Pase transcript levels were similarly affected in vivo in liver and in hepatocytes from LKO mice. For example, PEPCK transcripts were significantly (~70%) decreased in LKO hepatocytes, but were only reduced by ~20% in vivo, and this effect was not statistically significant. This more limited effect should be acknowledged, and not buried as a supplemental figure. It also could be mentioned in discussion.

Thank you, this discrepancy is now clearly described in the results section at the end of the first paragraph on page 8 and at the end of the first paragraph on page 16 of the discussion.

13. p. 8, line 179. Should be “glucagon” instead of “glycogen”.

Corrected, thank you for pointing out.

14. p. 7-8, lines 178-179. The point that basal glucose production is reduced in LKO, but the ability of glucagon to stimulate (fold-increase) glucose production is intact can, and should be stated more clearly. This is an important point, and warrants an additional bar graph to show that glucagon stimulated fold-increase is not dependent on GCN5L1, in my opinion.

Thank you for pointing this out. This is now included as the new panel (Fig 5c) and is discussed in the text, in the results section in the last sentence on page 8 and extending onto page 9. The new panel is shown below.

15. p. 8, lines 199-200. The data shown in Supplementary Figs 4b and 4c is important since it provides direct evidence that mitochondrial *GCN5L1* is positively linked to gluconeogenic gene expression glucose production, and would be best shown within Fig 6.

The figures and text have been reworked to streamline the data. Figure 4 now introduces the finding of reduced gluconeogenesis in the LKO hepatocytes and shows that the mitochondrial targeted adenoviral construct is localized to mitochondria and that this restores gluconeogenic transcript levels. Reconstitution with either the wildtype and mitochondrial targeted *GCN5L1* is then used in all results sections from Fig 6 through 8 to show that the rescue to *GCN5L1* restores FoxO1 levels (Fig 6), blunts ERK phosphorylation (Fig 7), blunts mitochondrial ROS and restores glucose production in the concluding Fig 8.

16. p. 8, line 199 and Fig 6b. The organization of the figures in Fig 6 is confusing, with 6b located below 6a, and then 6c located on the same level of 6a. I suggest showing 6b, together with the data in supplemental figures 4a and 4b, before showing studies related to FoxO1 turnover.

This figure has been reconfigured to improve the flow. Supplemental Figure 4 has now been incorporated into the main figures as described above (point 15) to improve the flow of the text. Thank you.

17. p. 9, line 203-205. The data presented indicate that Mt-*GCN5L1* plays an important role in promoting or maintaining *PEPCK* and *G6Pase* expression and glucose production in hepatocytes – but this is not the same as “regulation”, e.g. in response to fasting or feeding.

We agree with this and the additional data added to the manuscript focus on the retrograde signaling in the regulation of FoxO1 protein stability and not on the fasting regulation.

18. p. 9, line 216. It is important to know what level of FoxO1 protein was expressed. If FoxO1 expression is more than recovered and it is markedly overexpressed, then the recovery of gene expression and glucose production may reflect an over compensation due to very high levels of Foxo1. Put another way, this recovery experiment does not rule out the possibility that other factors also may contribute to reduced PEPCCK and G6Pase expression and glucose production in LKO cells, and this should be noted in the text and/or discussion.

Thank you, a sentence has been added to the final sentence of the first paragraph on page 10 that points out that FoxO1 overexpression may have additional effects on gluconeogenesis independent on the GCN5L1 signaling effects.

19. p. 9, line 224. Should read “differentially” instead of “differentiated”.

Corrected.

20. p. 20. Line 247. Suggest ERK “dependent” instead of “mediated”.

Corrected

21. p. 12, lines 288-299. Awkward. “although” occurs twice in this sentence.

This is now on page 14, in the second paragraph and both sentences have been restructured and both the ‘although’s’ have been deleted. Thank you.

22. p. 13. Line 294. Suggest - Insert “with” after “in contrast”.

Corrected, thank you.

23. p. 13, line 310. Suggest “did” instead of “was”.

Corrected.

Reviewer #2 (Remarks to the Author):

The authors have satisfactorily addressed my previous concerns. Their focus on GCN5L1 and one mechanism by which its ablation controls FOXO1 stability, along with their removal of the sirtuin data, together serve to focus the manuscript. Their initial characterization of liver-specific GCN5L1KO mice is interesting, and stimulates several future studies.

Reviewer #3 (Remarks to the Author):

In this revised manuscript, the authors added more mechanism by which hepatic depletion of GCN5L1 affects Foxo1 activity and gluconeogenesis. They now suggested that increased Erk activity was observed, and is the key regulator for the metabolic phenotype in GCN5L1 liver-specific KO mice. They also did not find any changes in mitochondrial function, so they don't have any clue how the depletion of GCN5L1 can affect cytosolic signaling and, as a result, hepatic gluconeogenesis. Without the clear-cut molecular mechanism linking any changes in mitochondrial events (that are direct targets of GCN5L1) and ERK-Foxo1 pathway, this study is more of a speculation and a description of their findings in non-physiological setting. As shown in their response to reviewer 1's question, the authors also admitted that GCN5L1 can no longer be considered as a nutrient sensor since its expression pattern does not match with Foxo1 activity. This significantly dampens the reviewer's enthusiasm to recommend the manuscript for publication.

Thank you for raising this concern. Our statement that there was no change in mitochondrial function was not sufficiently explicit given that we did only assessed limited mitochondrial functions.

As hydrogen peroxide levels have been found to regulate ERK phosphorylation, we extended our studies to explore if H₂O₂ levels were modulated by the genetic KO of GCN5L1. The measurement of H₂O₂ in primary hepatocytes and in isolated mitochondria showed higher H₂O₂ levels in LKO cells and mitochondria. These are now shown as new figures 8a, b. and shown below with the additional demonstration that antioxidants DPI and mito TEMPO can blunt the GCN5L1 hepatocyte KO effect.

In addition, the attenuation of ROS levels by DPI and mito TEMPO both blunted the excess ERK phosphorylation in GCN5L1 KO hepatocytes. Furthermore, in the presence of mito TEMPO GCN5L1 LKO primary hepatocytes showed a partial reversal of the blunted expression of PEPCK and G6Pase and the restoration of glucose production, which again supports the role of this signaling pathway in the

reduction of gluconeogenesis in the GCN5L1 KO hepatocytes. These new data are now shown in new figures 8 c, d and e and shown below.

Finally, to genetically confirm these findings, primary LKO hepatocytes were induced with adenoviral vectors to restore whole cell or mitochondrial GCN5L1. Here too, we show that the reconstitution of GCN5L1 blunts mitochondrial ROS levels and reversed the suppressed glucose production. These are shown as the new Figures 8 f and g and the relocation of figure 8h. Of note the mt-GCN5L1 is the adenoviral construct that is targeted to mitochondria. The panels are shown below:

As to your question about the nutrient sensing role of GCN5L1 we do find that GCN5L1 levels are significantly induced in wildtype mice at 6 hours of fasting. This is the time point where we see the maximal differences in glucose levels comparing the WT and GCN5L1 KO mice. This is discussed in the results section and shown in Supplementary Figure 2a and shown here.

This concept is also included in the discussion section on page 13, line 15 onwards. Where we point out that this is an interesting observation, that needs to be explored further.

We would propose that this additional data does now identify a mitochondrial mediated signaling event in the modulation of gluconeogenesis in our mouse model. Furthermore, this work expands our understanding of the role of mitochondrial retrograde signaling in the regulation of an important gluconeogenic transcription factor FoxO1. We also have identified an important role of ERK activation in the control of FoxO1 levels and have identified additional evidence linking mitochondrial ROS to ERK signaling.

We do agree that additional data and experiments will need to be performed to understand this in-vivo link between the nutrient dependent flux in GCN5L1 and gluconeogenesis but believe that this will need to be undertaken using a substantial number of additional experiments in in-vivo models and that is beyond the scope of this study that explores retrograde mitochondrial signaling.

Reviewers' Comments:

Reviewer #1:

Remarks to the Author:

The authors have responded to each of my concerns and suggestions.

Reviewer #3:

Remarks to the Author:

The revised manuscript satisfactorily addressed all the points raised by the reviewers.